# The E3 ubiquitin ligase mindbomb1 controls planar cell polarity-dependent convergent extension movements during zebrafish gastrulation

Vishnu Muraleedharan Saraswathy, Akshai Janardhana Kurup, Priyanka Sharma, Sophie Polès, Morgane Poulain, Maximilian Fürthauer*

Université Côte d'Azur, CNRS, Inserm, iBV, Nice, France

**Abstract** Vertebrate Delta/Notch signaling involves multiple ligands, receptors and transcription factors. Delta endocytosis – a critical event for Notch activation – is however essentially controlled by the E3 Ubiquitin ligase Mindbomb1 (Mib1). Mib1 inactivation is therefore often used to inhibit Notch signaling. However, recent findings indicate that Mib1 function extends beyond the Notch pathway. We report a novel Notch-independent role of Mib1 in zebrafish gastrulation. *mib1* null mutants and morphants display impaired Convergence Extension (CE) movements. Comparison of different *mib1* mutants and functional rescue experiments indicate that Mib1 controls CE independently of Notch. Mib1-dependent CE defects can be rescued using the Planar Cell Polarity (PCP) downstream mediator RhoA, or enhanced through knock-down of the PCP ligand Wnt5b. Mib1 regulates CE through its RING Finger domains that have been implicated in substrate ubiquitination, suggesting that Mib1 may control PCP protein trafficking. Accordingly, we show that Mib1 controls the endocytosis of the PCP component Ryk and that Ryk internalization is required for CE. Numerous morphogenetic processes involve both Notch and PCP signaling. Our observation that during zebrafish gastrulation Mib1 exerts a Notch-independent control of PCP-dependent CE movements suggest that Mib1 loss-of-function phenotypes should be cautiously interpreted depending on the biological context.

*For correspondence: furthauer@unice.fr

Competing interest: The authors declare that no competing interests exist.

## Introduction

Endocytic membrane trafficking is essential to control the abundance, localization, and activity of cellular signaling molecules. Depending on the biological context, the internalization of proteins from the cell surface allows desensitization to extracellular stimuli, formation of endosomal signaling compartments, re-secretion of signaling molecules through endosomal recycling or their lysosomal degradation (*Hupalowska and Miaczynska, 2012*; *Villaseñor et al., 2016*). One key example for the importance of endosomal membrane trafficking is provided by the Delta/Notch signaling pathway, where Delta ligand endocytosis is required for Notch receptor activation (*Chitnis, 2006*; *Fürthauer and González-Gaitán, 2009*; *Le Borgne et al., 2005a*; *Seib and Klein, 2021*).

Notch receptors are single-pass transmembrane proteins that interact with Delta/Serrate/Lag2 (DSL) family ligands (*Bray, 2016*; *Hori et al., 2013*). Productive ligand/receptor interactions trigger a series of proteolytic cleavages that release the Notch Intracellular Domain (NICD) into the cytoplasm, allowing it to enter the nucleus and associate with transcriptional cofactors to activate target gene expression. Studies in *Drosophila* and vertebrates revealed that Ubiquitin-dependent endocytosis of DSL ligands in signal-sending cells is essential to promote Notch receptor activation in adjacent

**eLife digest** Animal embryonic development involves producing an entire animal from a single starting cell, the zygote. To do this, the zygote must divide to make new cells, and these cells have to arrange themselves into the correct body shape. This requires a lot of cells to move in a coordinated fashion. One of these movements is called 'convergent extension', in which a typically round group of cells rearranges into a long, thin shape, for example, to increase the distance between the head and the tail of the animal. In order to coordinate this movement, cells need to communicate with each other. One of the signaling pathways cells use to guide them to the right positions is the planar cell polarity (PCP) pathway.

Zebrafish are used to study PCP in convergent extension because they are transparent, making it easy to track their cell movements under the microscope. Interestingly, when a protein called Mindbomb1 (Mib1) is inactivated in zebrafish embryos, convergent extension is reduced. Mib1 helps control the activity of other proteins by attaching a chemical marker called ubiquitin to them, which tags these proteins to be relocated from the cell surface to small vesicles within the cell. The protein is known to be involved in the formation of neurons – the cells that make up the brain and nerves – but its links to cell movement and the PCP pathway had not been explored.

Saraswathy et al. used a technique called Crispr/Cas9 mutagenesis to genetically modify zebrafish and then used observations under the microscope to determine the role of Mib1 in PCP and convergent extension. Their experiments show that Mib1 helps internalize a protein called Ryk from the cell surface into the cell. This internalization of Ryk is required to relay signals through the PCP pathway. When Mib1 is missing, Ryk stays on the surface of the cell, instead of moving to the inside, blocking PCP signaling between cells and therefore blocking convergent extension.

Understanding the role of Mib1 in PCP signaling sheds light on how cell movements are coordinated during the embryonic development of zebrafish. Future research will involve determining whether Mib1 plays the same role in other animals, offering further insights into embryonic development. Additionally, PCP is known to have a role in disease, including the spread of cancer. It will be important to determine whether Mib1 is involved in this process as well.

signal-receiving cells (*Deblandre et al., 2001*; *Itoh et al., 2003*; *Lai et al., 2001*; *Le Borgne and Schweisguth, 2003*; *Pavlopoulos et al., 2001*).

While the precise mechanism through which Delta promotes Notch activation is still under investigation, current models suggest that the endocytosis of DSL ligands represents a force-generating event that physically pulls on Notch receptors to promote their activation (*Langridge and Struhl, 2017*; *Meloty-Kapella et al., 2012*; *Seib and Klein, 2021*). Notch pathway activation is therefore critically dependent on Delta endocytosis, a process controlled by ligand polyubiquitination. Protein ubiquitination involves ubiquitin-activating E1 enzymes, ubiquitin-conjugating E2 enzymes and substrate-specific E3 ubiquitin ligases (*Oh et al., 2018*). Through their ability to recognize specific substrates, different E3 ligases control the activity of various cellular signaling pathways.

Genetic studies revealed that two different RING (Really Interesting New Gene) finger domain E3 ligases, Neuralized (Neur) and Mindbomb (Mib) control DSL ligand endocytosis in *Drosophila* (*Lai et al., 2001*; *Le Borgne et al., 2005b*; *Le Borgne and Schweisguth, 2003*; *Pavlopoulos et al., 2001*). Vertebrate genomes harbor two *mib* and several *neur* homologues. However, mouse *neur1* or *neur2* single and double mutants present no phenotypes indicative of defective Notch signaling (*Koo et al., 2007*). In contrast, Notch signaling is severely impaired upon genetic inactivation or morpholino knock-down of *mib1* in mice, *Xenopus* and zebrafish (*Itoh et al., 2003*; *Koo et al., 2007*; *Yoon et al., 2008*). In addition to Mib1, its orthologue Mib2 as well as Asb11 (a component of a multi-subunit Cullin E3 ligase complex) have been implicated in DSL ligand endocytosis (*Sartori da Silva et al., 2010*; *Zhang et al., 2007a*). Nonetheless, mutational analysis in zebrafish failed to confirm a requirement for *mib2* in Notch signaling (*Mikami et al., 2015*) and showed that a *mib1* inactivation is sufficient to essentially abolish the expression of a transgenic Notch reporter in the central nervous system (*Sharma et al., 2019*). These findings identify Mib1 as the major regulator of vertebrate DSL ligand endocytosis.

In addition to Delta ligands, Mib1 is able to interact with a number of additional substrate proteins (*Berndt et al., 2011*; *Matsuda et al., 2016*; *Mertz et al., 2015*; *Tseng et al., 2014*). Accordingly, functional studies have implicated Mib1 in a growing number of functions that include the regulation of epithelial morphogenesis (*Dho et al., 2019*; *Matsuda et al., 2016*) and cell migration (*Mizoguchi et al., 2017*), centrosome and cilia biogenesis (*Douanne et al., 2019*; *Joachim et al., 2017*; *Villumsen et al., 2013*; *Wang et al., 2016*; *Cajanek et al., 2015*), the control of glutamate receptor localization (*Sturgeon et al., 2016*) or interferon production (*Li et al., 2011*).

A study in human cell culture identified the Receptor-like tyrosine kinase Ryk as a target of Mib1-mediated ubiquitination (*Berndt et al., 2011*). Ryk is a single-pass transmembrane protein that binds Wnt ligands through its extracellular Wnt Inhibitory Factor (WIF) domain. While its intracellular pseu-dokinase domain appears devoid of functional enzymatic activity, Ryk has been suggested to regu-late cell signaling through scaffolding functions or the γ-secretase-dependent release and nuclear translocation of its intracellular domain (*Roy et al., 2018*). A number of studies have implicated Ryk in canonical, β-catenin-dependent Wnt signaling (*Green et al., 2008*; *Lu et al., 2004*; *Roy et al., 2018*). In this context, Mib1 has been shown to promote the ubiquitination and internalization of Ryk, which appears to be required for Wnt3A-mediated β-catenin stabilization/activation (*Berndt et al., 2011*). While experiments in *C. elegans* provided evidence for genetic interactions between *mib1* and *ryk* (*Berndt et al., 2011*), the importance of Mib1/Ryk interactions for vertebrate development or physi-ology has not been addressed.

In addition to its role in canonical Wnt signaling, several studies have linked Ryk to the non-canonical, β-catenin independent, Wnt/Planar Cell Polarity (PCP) and Wnt/Ca²⁺ pathways (*Duan et al., 2017*; *Kim et al., 2008*; *Lin et al., 2010*; *Macheda et al., 2012*; *Roy et al., 2018*). *Ryk* mutant mice present a range of diagnostic PCP phenotypes, including defects in neural tube closure and the orien-tation of inner ear sensory hair cells (*Andre et al., 2012*; *Macheda et al., 2012*). During early fish and frog development, PCP signaling regulates Wnt-dependent Convergent Extension (CE) movements that direct embryonic axis extension (*Butler and Wallingford, 2017*; *Davey and Moens, 2017*; *Gray et al., 2011*; *Tada and Heisenberg, 2012*). Different studies have suggested that Ryk may control PCP by acting together with Wnt-binding Frizzled (Fz) receptors (*Kim et al., 2008*), regulating a Fz-in-dependent parallel pathway (*Lin et al., 2010*) or by controlling the stability of the core PCP pathway component Vangl2 (*Andre et al., 2012*).

In both vertebrates and invertebrates, the core PCP machinery is defined by three transmembrane proteins Flamingo(Fmi)/CELSR, Fz and Strabismus/Vangl as well as their cytoplasmic partners Dishev-elled (Dvl), Prickle (Pk) and Diego (Dgo)/ANKRD6 (*Butler and Wallingford, 2017*; *Devenport, 2014*; *Harrison et al., 2020*; *Humphries and Mlodzik, 2018*; *Vladar et al., 2009*). Polarity is established through formation of distinct CELSR/Vangl/Pk and CELSR/Fz/Dvl/Dgo complexes at the opposite sides of the cell. The fact that similar defects are often observed upon overexpression or inactiva-tion of PCP pathway components suggests that the levels of individual proteins need to be tightly controlled. It is therefore no surprise that factors such as Dynamin, which governs endocytic vesicle scission, the early endosomal GTPase Rab5 and other regulators of membrane trafficking have been shown to control the distribution of the transmembrane proteins Fmi/CELSR, Fz and Vangl2 (*Butler and Wallingford, 2017*; *Devenport et al., 2011*; *Mottola et al., 2010*; *Strutt and Strutt, 2008*). In addition to regulating the trafficking of core pathway components, endocytosis ensures the PCP-dependent regulation of cellular adhesion molecules (*Classen et al., 2005*; *Ulrich et al., 2005*).

While numerous studies therefore indicate a central role of membrane trafficking in PCP, it remains to be established whether Ryk/PCP signaling is subject to endo-lysosomal control. In the present study, we show that Mib1-mediated Ryk endocytosis is required for the PCP-dependent control of CE movements during zebrafish gastrulation. The analysis of different *mib1* mutant alleles shows that the role of Mib1 in PCP is separable from its function in Notch signaling. Our work thereby identifies a novel function of this E3 ubiquitin ligase in the control of PCP-dependent morphogenetic movements.

## Results

### Mindbomb1 regulates convergent extension independently of Notch

Through its ability to promote Delta ligand endocytosis, the E3 Ubiquitin ligase Mindbomb1 plays an essential role in vertebrate Notch receptor activation (*Guo et al., 2016*). In the course of experiments

that were initially designed to study the role of Notch signaling in the morphogenesis of the zebrafish nervous system (*Sharma et al., 2019*), we realized that embryos injected with a mib1 morpholino (mib1 morphants) present a reduced axial extension at the end of gastrulation that is indicative of defects in embryonic Convergence Extension (CE) movements (*Figure 1A*, *Figure 1—figure supplement 1A, B*). Accordingly, mib1 morphants present a widening of the notochord, somites, and neural plate (*Figure 1B*, *Figure 1—figure supplement 1C*). The mib1 exon/intron1 splice site morpholino used in these experiments has been previously validated in different studies (*Itoh et al., 2003*; *Sharma et al., 2019*). We further confirmed its specificity by showing that the co-injection of a WT mib1 RNA that is not targetable by the morpholino restores axis extension (*Figure 1A*).

The *mib1^ta52b* mutation in the C-terminal Mib1 RING finger domain (RF3, *Figure 1C*) disrupts the ability of the protein to promote Delta ubiquitination (*Itoh et al., 2003*; *Sharma et al., 2019*; *Zhang et al., 2007a*). The Mib1^ta52b mutant protein retains however the ability to bind Delta ligands (*Itoh et al., 2003*; *Zhang et al., 2007a*). Through this ability to sequester Delta in an enzymatically inactive complex, Mib1^ta52b exerts a dominant-negative activity that leads to the appearance of stronger Notch loss of function phenotypes in *mib1^ta52b* point mutants compared to *mib1^tfi91* null mutants (*Mikami et al., 2015*; *Zhang et al., 2007b*). In spite of the strong, antimorphic Notch loss-of-function phenotypes observed in *mib1^ta52b* mutants, axial extension occurs normally in these animals (*Figure 1D*, *Figure 1—figure supplement 2A*). This observation raises the question whether Mib1 exerts a Notch-independent function in CE. In accordance with this hypothesis, a constitutively activated form of Notch (NICD) that is able to restore Notch-dependent defects in the nervous system (*Sharma et al., 2019*) fails to rescue mib1 morphant axis extension (*Figure 1E*, *Figure 1—figure supplement 3A*).

Sequencing of the *mib1* cDNA in mib1 morphants revealed a retention of intron1 that causes the appearance of a premature Stop codon (*Figure 1—figure supplement 3B*). As a consequence, the Mib1 morphant protein comprises only the first 76 amino acids of WT Mib1 (*Figure 1C*). We hypothesized that this early termination of the open-reading frame could disrupt functions that are not affected by the *mib1^ta52b* point mutation. Accordingly, mib1 WT and *mib1^ta52b* mutant RNAs are equally capable of rescuing mib1 morphant CE phenotypes (*Figure 1A and F*, *Figure 1—figure supplement 3C*).

As *mib1^ta52b* point mutants show normal CE, we further studied axis extension in *mib1* null mutants. The previously reported *mib1^tfi91* allele causes a truncation of the Mib1 open-reading frame after 59 amino acids and therefore likely represents a molecular null (*Figure 1C*; *Itoh et al., 2003*). In contrast to *mib1^ta52b* mutants, *mib1^tfi91* homozygous animals present CE defects that are statistically significant, although weaker than in mib1 morphants (*Figure 1G and H*, Cohen's d effect size = 0.49 for *mib1^tfi91*, 1.23 for MO mib1). Similar phenotypes are observed for a new potential null allele generated in the present study (*mib1^nce2a*, *Figure 1C and G*, *Figure 1—figure supplement 2C, F*) or in *mib1^tfi91/nce2a* trans-heterozygotes (*Figure 1G*). The *mib1^tfi91* and *mib1^nce2a* mutations introduce stop codons shortly after the beginning of the *mib1* open-reading frame (*Itoh et al., 2003* and *Figure 1—figure supplement 2F*), a mutation pattern that can cause nonsense mediated decay of mutant mRNAs and could thereby trigger partial transcriptional compensation (*El-Brolosy et al., 2019*). Accordingly, *mib1* transcript levels appear reduced in *mib1^tfi91* and *mib1^nce2a* but not in *mib1^ta52b* mutants (*Figure 1I*, *Figure 1—figure supplement 2B, D,E*).

Zebrafish Mib1 interacts with Epb41l5 to regulate neuronal differentiation (*Matsuda et al., 2016*) and with Catenin delta1 to control cell migration (*Mizoguchi et al., 2017*). Both of these activities are disrupted in *mib1^ta52b* mutants (*Matsuda et al., 2016*; *Mizoguchi et al., 2017*). Our observation that mib1 morphants (*Figure 1A*) or *mib1* null mutants (*Figure 1G*) but not *mib1^ta52b* mutants (*Figure 1D*) present defects in gastrulation stage axial extension identify thereby a novel role of Mib1 in the regulation of zebrafish CE movements.

## Mindbomb1 RING finger domains are required for convergent extension movements

Vertebrate CE requires non-canonical Wnt/PCP signaling (*Butler and Wallingford, 2017*; *Davey and Moens, 2017*; *Gray et al., 2011*; *Tada and Heisenberg, 2012*). To test whether Mib1 loss-of-function impairs PCP pathway activity in gastrulating zebrafish embryos, we overexpressed the PCP downstream effector RhoA in mib1 morphants. RhoA fully restores axis extension (*Figure 2A*), suggesting thereby that Mib1 is required for the PCP-dependent control of embryonic CE movements.

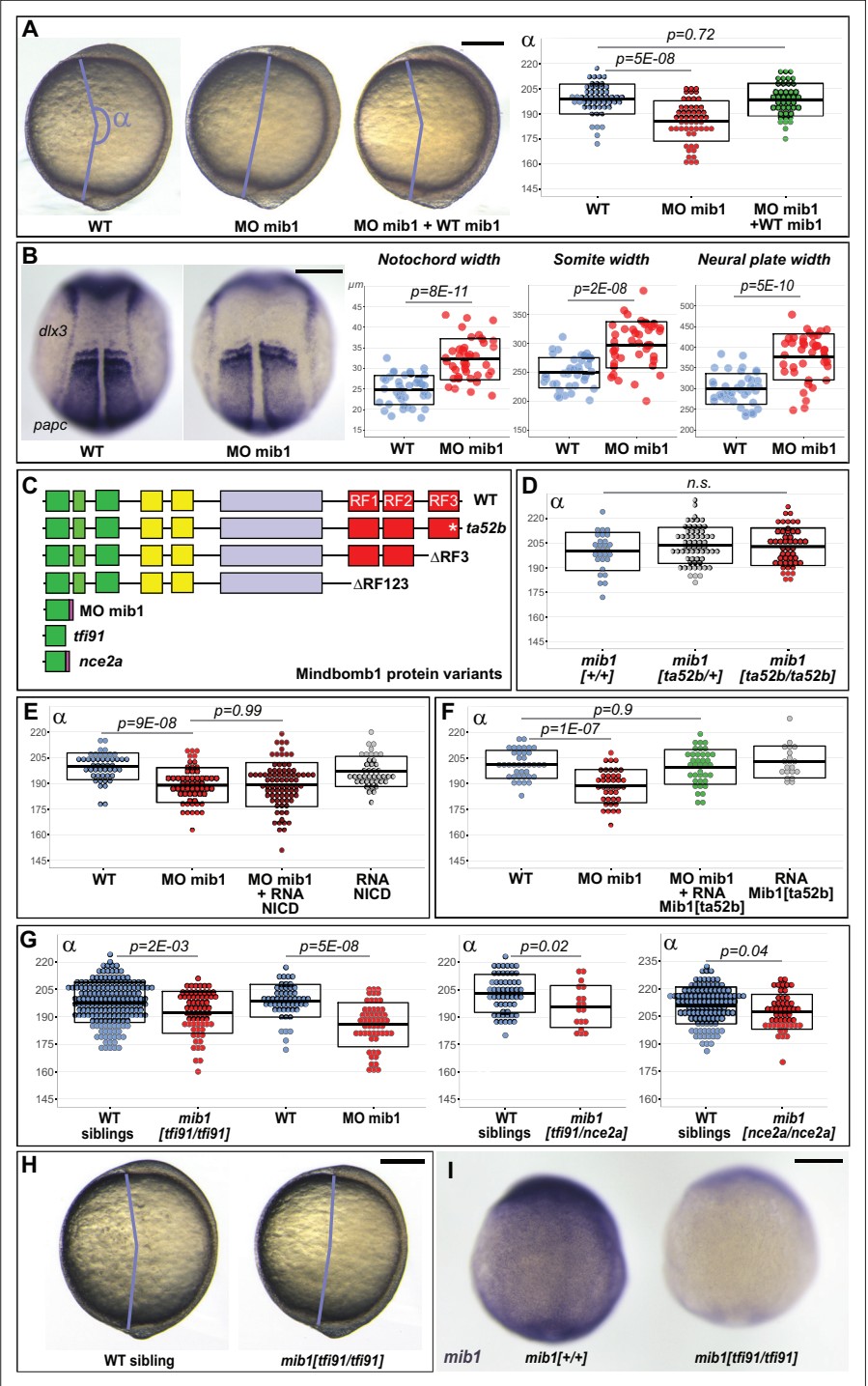

**Figure 1.** Mib1 regulates PCP-dependent convergent extension movements independently of Notch. (**A**) Axis extension was quantified at bud stage by measuring the axis extension angle α. Axis extension is reduced in mib1 morphants but restored upon coinjection of WT mib1 RNA. Lateral views of bud stage embryos, anterior up, dorsal to the right. (**B**) mib1 morphants present a widening of the notochord, somites, and neural plate. Dorsal views of 2 somite stage embryos, anterior up. *dlx3* in situ hybridization outlines the neural plate, *papc* the somites and the adaxial cells lining the notochord. Widths indicated in microns. (**C**) Mib1 protein variants used in the study. (**D**) The *mib1^ta52b* mutation has no effect on axis extension. (**E**) Constitutively activated Notch (NICD) fails to restore mib1 morphant axis extension. (**F**) mib1^ta52b RNA injection restores mib1 morphant axis extension. (**G,H**) Axis extension is impaired in *mib1^tfi91* or *mib1^nce2a* null mutants. On the left panel of (**G**) the mib1 morphant data from (**A**) are included for comparison. (**I**) In situ hybridization reveals reduced *mib1* transcript levels in n = 27 *mib1^tfi91* mutant

*Figure 1 continued on next page*

*Figure 1 continued*

embryos. Dorsal views of bud stage embryos, anterior up. To warrant identical acquisition conditions, two embryos were photographed on a single picture. Scalebars: 200 µm. Boxes in (**A,B, D–G**) represent mean values ± SD. See *Figure 1—source data 1* for complete statistical information.

The online version of this article includes the following source data and figure supplement(s) for figure 1:

**Source data 1.** Complete statistical information for the experiments reported in *Figure 1* and *Figure 1—figure supplement 1*.

**Figure supplement 1.** Mib1 knock-down impairs axial elongation.

**Figure supplement 2.** Axis extension and RNA expression in different *mib1* mutants.

**Figure supplement 3.** Mib1 regulates convergent extension independently of Notch.

All Mib1 functions known to date require its E3 ubiquitin ligase activity that is dependent of the presence of C-terminal RING finger domains (*Guo et al., 2016*). In contrast, the N-terminal part of the protein is responsible for the interaction of Mib1 with different substrates (*Berndt et al., 2011*; *Itoh et al., 2003*; *Zhang et al., 2007a*). In the context of Delta/Notch signaling, truncated Mib1 variants that lack all three RING Finger domains sequester Delta ligands without promoting their ubiquitination and thereby exert a dominant-negative effect. Similarly, a truncated Mib1 variant that lacks all three RING Finger domains (Mib1$^{\Delta RF123}$, *Figure 1C*) enhances the defects of mib1 morphants as well as impairing axis extension in WT animals (*Figure 2B*, *Figure 1—figure supplement 1*). The enhanced CE defects of Mib1$^{\Delta RF123}$-injected mib1 morphants are likely due to its capacity to interfere with maternally provided Mib1 protein which is unaffected by our morpholino that only impairs the splicing of

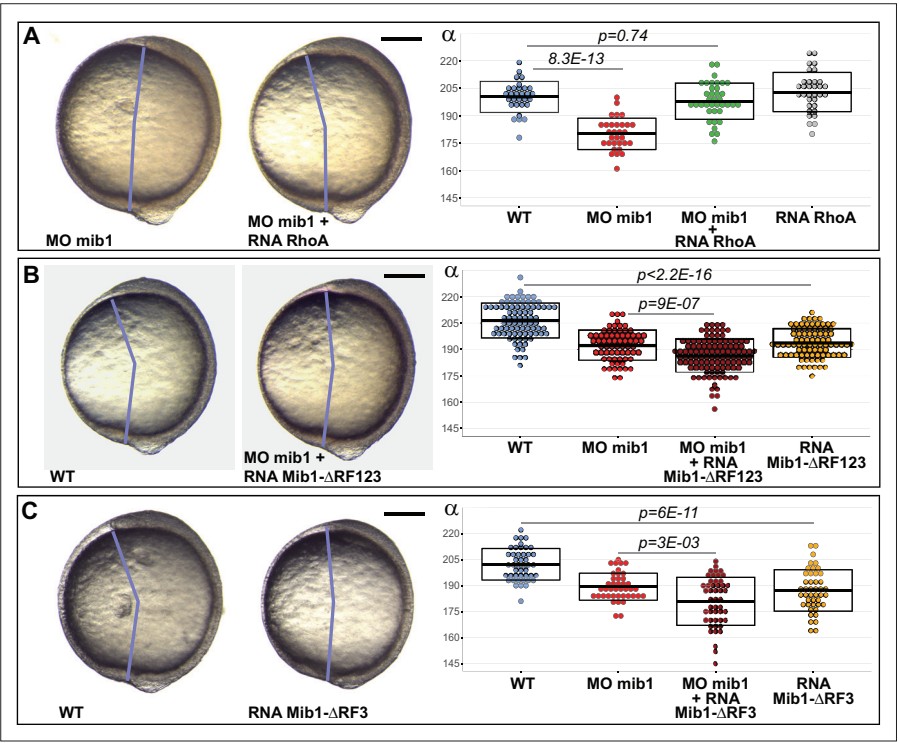

**Figure 2.** Mib1 controls PCP through its RING finger domains. (**A**) RhoA overexpression rescues mib1 morphant axis extension. (**B,C**) Mib1 proteins lacking all (Mib1ΔRF123, **B**) or only the last (Mib1ΔRF3, **C**) RING finger impair axis extension in mib1 morphant or WT embryos. Lateral views of bud stage embryos, anterior up, dorsal to the right. Scalebars: 200 µm. Boxes represent mean values ± SD. *Figure 2—source data 1* for complete statistical information.

The online version of this article includes the following source data for figure 2:

**Source data 1.** Complete statistical information for the experiments reported in *Figure 2*.

zygotically produced mib1 transcripts. A Mib1 variant lacking only the last RING finger (Mib1ΔRF3, *Figure 1C*) produced similar results (*Figure 2C*, *Figure 1—figure supplement 1*).

Our results suggest that the substrate-ubiquitinating Mib1 RING-finger domains are required for PCP. As Ubiquitin-dependent membrane trafficking is important for PCP (*Butler and Wallingford, 2017*; *Devenport, 2014*; *Feng et al., 2021*), we set out to determine whether Mib1 controls CE by regulating the trafficking of a PCP pathway component.

## Convergent extension requires Mindbomb1-dependent Ryk internalization

Mammalian Mib1 has been shown to control the ubiquitin-dependent endocytic internalization of the the Wnt co-receptor Receptor like tyrosine kinase Ryk (*Berndt et al., 2011*). Interestingly, studies in mice, frogs and zebrafish have implicated Ryk in non-canonical Wnt/PCP signaling (*Kim et al., 2008*; *Lin et al., 2010*; *Macheda et al., 2012*). To determine whether Mib1 regulates CE by controlling Ryk internalization, we started by analyzing the effect of Mib1 gain of function on Ryk localization. Ryk localizes to the cell surface as well as intracellular compartments (*Figure 3A*), 70.7% of which are positive for the early endosomal marker Rab5 (n = 75 cells from eight embryos, *Figure 3—figure supplement 1A*; *Berndt et al., 2011*; *Kim et al., 2008*; *Lin et al., 2010*). In accordance with a role of Mib1 in promoting Ryk endocytosis, Mib1 overexpression depleted Ryk from the cell cortex and triggered its accumulation in intracellular compartments (*Figure 3B*) without affecting the localization of the general plasma membrane marker GAP43-RFP (*Figure 3—figure supplement 1B, C*).

If Mib1-dependent Ryk endocytosis is important for CE, this process should be unaffected by the *mib1*^ta52b mutation that disrupts Notch signaling (*Itoh et al., 2003*) but has no effect on gastrulation movements (*Figure 1D*). In accordance with this hypothesis, Mib^ta52b misexpression promotes a relocalization of Ryk from the plasma membrane toward intracellular compartments (*Figure 3—figure supplement 2A, B*), similar to the effect observed upon overexpression of wild-type Mib1 (*Figure 3—figure supplement 1B, C*).

To determine whether Mib1 specifically affects Ryk or acts as a general regulator of PCP protein trafficking, we tested the effect of Mib1 overexpression on the localization of different transmembrane proteins. Ryk has been shown to interact with the core PCP component Vangl2 whose endocytic trafficking is crucial for PCP (*Andre et al., 2012*). In contrast to Ryk, Mib1 overexpression has no obvious effect on Vangl2 localization (*Figure 3C and D*). Similarly, we detected no impact of Mib1 overexpression on the localization of Fz2 and Fz7, two Wnt receptors that have been in implicated in PCP signaling (*Kim et al., 2008*; *Lin et al., 2010*; *Capek et al., 2019*; *Figure 3—figure supplement 3A-D*). Ryk belongs to a family of Wnt-binding Receptor Tyrosine Kinases that have the particularity of harboring a most likely inactive pseudokinase domain (*Roy et al., 2018*). Other members of this protein family include the Receptor tyrosine kinase-like Orphan Receptors (ROR), among which zebrafish ROR2 has been implicated in CE (*Bai et al., 2014*; *Mattes et al., 2018*). Our experiments failed to identify an effect of Mib1 overexpression on the localization of zebrafish ROR2 or its orthologue ROR1 (*Figure 3—figure supplement 3E-H*). Taken together, our observations suggest that Mib1 regulates CE movements through the specific control of Ryk localization.

To determine if Mib1 function is required for Ryk endocytosis, we quantified the number of RYK-positive endosomes in mib1 morphants and *mib1*^tfi91 null mutants. In these experiments, RNAs encoding RYK-GFP and Histone2B-mRFP were co-injected into mib1 morphant or *mib1*^tfi91 mutant embryos. The Histone2B-mRFP signal was used as an injection tracer to ascertain that the number of RYK-GFP-positive endosomes was scored in embryos that had received comparable amounts of injected material.

These experiments revealed a reduction in the number of Ryk-GFP positive endosomes in mib1 morphants (*Figure 3E*). Our observation that the CE defects of mib1 morphants can be further enhanced by the co-injection of antimorphic Mib1^ΔRF123 or Mib1^ΔRF3 constructs (*Figure 2B and C*) suggests that Mib1 activity is only partially reduced in mib1 morphants. Accordingly, increasing the amount of injected Ryk-GFP RNA from 3 to 12 pg restores the number of Ryk positive endosomes in mib1 morphants (*Figure 3E*, *Figure 3—figure supplement 4A, B*). Even a high dose of Ryk-GFP fails, however, to restore endosome number in embryos co-injected with mib1 morpholino and dominant-negative Mib1^ΔRF123 (*Figure 3E, F and G*, *Figure 3—figure supplement 4C, D*), suggesting that Ryk can no more be internalized once Mib1 function is severely compromised.

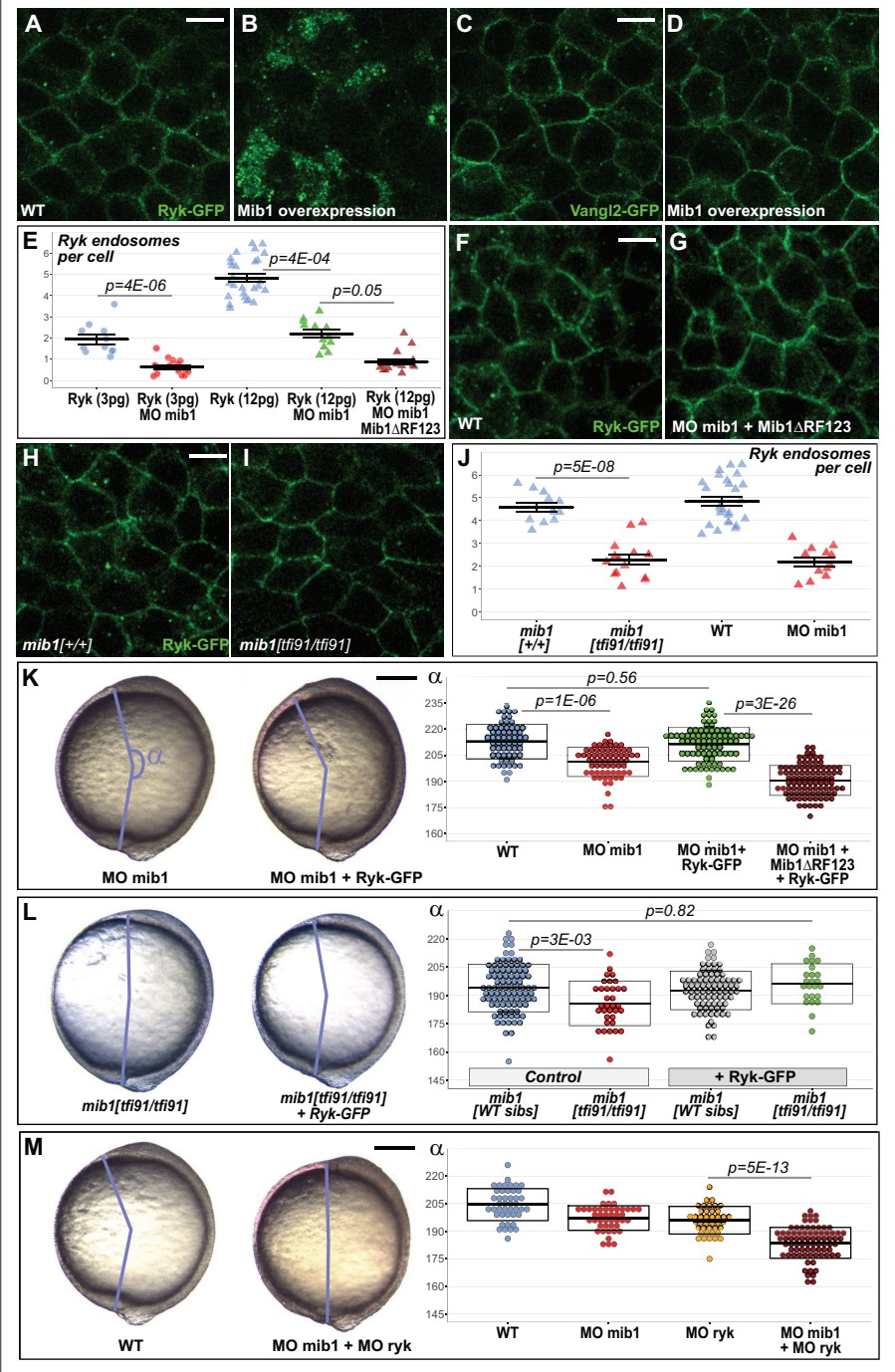

**Figure 3.** Mib1-mediated Ryk endocytosis controls Convergent Extension movements. (**A–D**) WT mib1 RNA injection triggers Ryk internalization in 20/21 embryos (**B**) but has no effect on Vangl2 localization (D, n = 23). (**E–G**) Mib1 morpholino injection reduces the number of Ryk endosomes that are present upon injection of Ryk-GFP RNA. Increasing the dose of Ryk-GFP RNA restores endosome number in mib1 morphants but not in embryos coinjected with Mib1ΔRF123. (**H–J**) The number of Ryk endosomes that are present upon injection of Ryk-GFP RNA (12 pg) is reduced in mib1 null mutants. mib1 morphant data from panel E are shown again for comparison. (**K**) Ryk-GFP RNA (12 pg) rescues axis extension in mib1 morphants but not in embryos coinjected with Mib1ΔRF123. (**L**) Similarly Ryk-GFP injection rescues axis extension in *mib1^{tf91}* mutants. (**M**) Ryk morpholino injection aggravates mib1 morphant axis extension phenotypes. (**A–D,F,G,H,I**) dorsal views of 90% epiboly stage embryos, anterior up, scalebars 10 μm. (**K,L,M**) Lateral views of bud stage embryos, anterior up, scalebars 200 μm. In (**E,J**) each data point represents the mean number of endosomes for 20 cells from a single embryo.

*Figure 3 continued on next page*

*Figure 3 continued*

For comparison J again includes the mib1 morphant from panel E. Bars represent mean values ± SEM. In (**K,L,M**) boxes represent mean values ± SD. See *Figure 3—source data 1* for complete statistical information.

The online version of this article includes the following source data and figure supplement(s) for figure 3:

**Source data 1.** Complete statistical information for the experiments reported in *Figure 3* and *Figure 3—figure supplement 5*.

**Figure supplement 1.** Mib1 promotes Ryk internalization and degradation.

**Figure supplement 2.** Mib1^ta52b overexpression promotes Ryk internalization.

**Figure supplement 3.** Mib1 overexpression does not affect Frizzled/Ror localization.

**Figure supplement 4.** Mib1 loss of function impairs Ryk endocytosis.

**Figure supplement 5.** mib1 morphant defects are not rescued upon vangl2 overexpression.

Similar to mib1 morphants, *mib1^tfi91* null mutants present a reduction in the number of RYK-GFP-positive endosomes (*Figure 3H, J*, *Figure 3—figure supplement 4E, F*). While stronger CE defects are observed in mib1 morphants compared to *mib1* mutants (*Figure 1G*), morphant and mutant embryos present a comparable reduction of the number of RYK-GFP-positive endosomes (*Figure 3J*, Cohen's d effect size = 3.02 for *mib1^tfi91*, 2.99 for mib1 morphants). These observations suggest the existence of a compensatory mechanism that allows to partially correct the PCP signaling defects that arise from a defect in Ryk endocytosis.

Why does impaired Ryk endocytosis cause CE defects? The phenotypes of Mib1-depleted embryos could be either due to the loss of Ryk-positive endosomal compartments, or to the accumulation of non-internalized Ryk at the cell surface. In *C. elegans,* Mib1 controls Ryk cell surface levels by promoting Ryk internalization and degradation (*Berndt et al., 2011*). Similarly, zebrafish embryos injected with a high level of Mib1 RNA present an overall loss of Ryk-GFP signal (*Figure 3—figure supplement 1F*) that can also be observed upon misexpression of the Mib1^ta52b mutant protein (*Figure 3—figure supplement 2C*). If the CE phenotypes of mib1 morphants are indeed due to increased Ryk cell surface levels, then Ryk-GFP overexpression should further enhance the defects of Mib1-depleted animals. The opposite prediction should, however, apply if the CE defects of mib1 morphants are due to the loss of Ryk-positive endosomal compartments. In this case Ryk-GFP overexpression, which allows to restore the number of Ryk endosomes in mib1 morphants (*Figure 3E*), should also rescue the CE phenotypes of Mib1-depleted animals. In accordance with this later hypothesis, Ryk-GFP over-expression restores axis extension in mib1 morphants (*Figure 3K*) and *mib1^tfi91* mutants (*Figure 3L*). In contrast, Vangl2 overexpression did not display rescuing activity (*Figure 3—figure supplement 5*).

If mib1 morphant CE defects are due to a loss of Ryk-positive endosomes, *ryk* knock-down is expected to further increase the severity of the observed phenotypes. To test this hypothesis, morpholinos directed against mib1 or ryk were injected separately or in combination. The injection of morpholino-insensitive mib1 or ryk RNAs allows to rescue the CE defects that are generated by their respective morpholinos, validating thereby the specificity of the reagents (*Figure 1A* and *Figure 4D*, *Figure 4—figure supplement 1A*). Co-injection of mib1 and ryk morpholinos causes CE defects that are significantly enhanced compared to single morphants (*Figure 3M*), adding further support to our hypothesis that the CE defect of Mib1-depleted embryos are due to Ryk loss of function.

## *Ryk* mutants are insensitive to *Mindbomb1* loss of function

The above-mentioned experiment shows that inhibiting Mib1 function enhances the CE defects of animals that present a partial loss of Ryk activity due to morpholino knock-down (*Figure 3M*). If Mib1 regulates CE by controlling Ryk endocytosis, Mib1 loss of function should, however, have no more enhancing effect in animals that are not only partially, but entirely devoid of Ryk activity. To test this hypothesis, we used Crispr/Cas9 mutagenesis to generate a stable *ryk* mutant line.

The use of a gRNA directed against exon 6 of zebrafish *ryk* led to the generation of *ryk^nce4g* mutants that present an 11 base pair insertion at the target locus. The presence of the mutation in *ryk^nce4g* transcripts was confirmed through sequencing of the complete *ryk* mutant cDNA (*Figure 4A*). The *ryk^nce4g* mutation introduces a premature stop codon that truncates the extracellular domain and deletes the transmembrane and cytoplasmic parts of the protein (*Figure 4A*). Accordingly, the *nce4g* mutation abolishes the detection of an HA-tag located at C-terminus of the wild-type protein (*Figure 4B and*

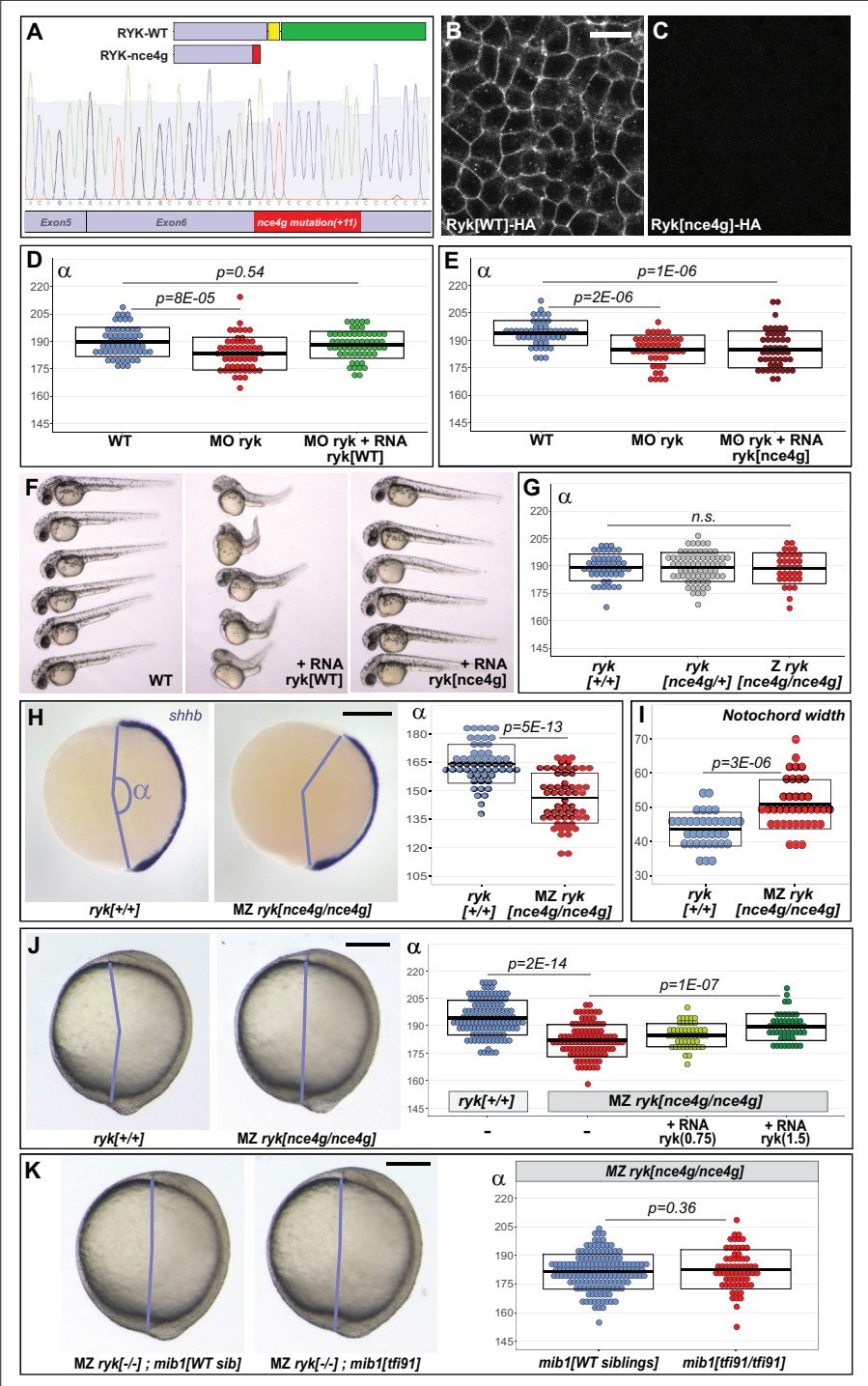

**Figure 4.** *mib1* loss of function has no effect on convergent extension in maternal zygotic *ryk* mutants.
(**A**) *ryk^nce4g^* mutants present an 11 base pair insertion in exon 6. The RYK-nce4g mutant protein comprises only
part of the extracellular (blue) and lacks the entire transmembrane (yellow) and intracellular (green) domains.
(**B,C**) Accordingly, a C-terminal HA tag that allows to localize WT Ryk (B, n = 12) becomes undetectable upon
introduction of the *ryk^nce4g^* mutation (C, n = 14). Dorsal views of 90% epiboly stage embryos, anterior up. Scalebar
20 μm. (**D,E**) The Convergent Extension (CE) phenotypes of ryk morphant animals can be rescued using 1.5 pg WT
ryk (**D**) but not ryk^nce4g^ mutant (**E**) RNA. (**F**) Overexpressing high levels (25 pg) WT ryk RNA causes severe embryonic
malformations while no effect is observed using ryk^nce4g^ mutant RNA. 32 hpf embryos, anterior to the left, dorsal
up (n = 24 embryos/condition). (**G**) Zygotic (Z) *ryk* loss of function does not impair CE. (**H–J**) In contrast, Maternal
Zygotic (MZ) *ryk* mutants present characteristic CE phenotypes such as a reduced axial elongation (H, *shhb* in situ

*Figure 4 continued on next page*

*Figure 4 continued*

hybridization) and an increased width of the notochord (I, *foxa3* in situ hybridization, see also **Figure 4—figure supplement 1F**). (**J**) ryk WT RNA injection allows a significant rescue of MZ *ryk* mutant CE defects. (**K**) Similar CE defects are observed in MZ *ryk* single mutants and MZ *ryk; mib1* double mutants. (**H,J,K**) Lateral views of bud stage embryos, anterior up, dorsal to the right. Scalebars 200 µm. In (**D,E,G–K**) boxes represent mean values ± SD. See **Figure 4—source data 1** for complete statistical information.

The online version of this article includes the following source data and figure supplement(s) for figure 4:

**Source data 1.** Complete statistical information for the experiments reported in **Figure 4** and **Figure 4—figure supplement 1**.

**Figure supplement 1.** Maternal zygotic *ryk^{nce4g}* mutants present Convergent Extension defects.

*C*). In contrast to wild-type RNA, *ryk^{nce4g}* RNA fails to rescue ryk morphant CE defects (**Figure 4D and E**, **Figure 4—figure supplement 1A, B**). While high levels of WT ryk RNA induce severe embryonic malformations, no overexpression phenotypes were observed using *ryk^{nce4g}* mutant RNA (**Figure 4F**).

Despite this evidence for functional *ryk* inactivation, our analysis failed to reveal CE defects in zygotic *ryk^{nce4g}* mutants (**Figure 4G**, **Figure 4—figure supplement 1C**). Frame-shift mutations can induce nonsense mediated degradation of mutant transcripts and thereby trigger a process of transcriptional adaptation that could compensate for the loss of function of the mutated protein (**El-Brolosy et al., 2019**). Accordingly, *ryk* transcript levels are reduced in *ryk^{nce4g}* mutants (**Figure 4—figure supplement 1D, G**).

Alternatively, the lack of CE phenotypes in zygotic *ryk^{nce4g}* mutants could be due to the persistence of maternally deposited *ryk* RNA or protein. *ryk^{nce4g}* mutants are viable and fertile and can thereby be used to generate embryos that are devoid of both maternal and zygotic Ryk function. In contrast to zygotic mutants, Maternal Zygotic (MZ) *ryk^{nce4g}* mutants present highly significant CE defects compared to wild-type control animals from the same genetic background (**Figure 4H–J**, **Figure 4—figure supplement 1E, F**, see methods for details). WT ryk RNA injection induces a significant rescue of CE, confirming thereby the specificity of the observed phenotypes (**Figure 4J**, **Figure 4—figure supplement 1H**). In further accordance with a loss of ryk activity in these animals, *ryk^{nce4g}* mutants are insensitive to the injection of a translation-blocking ryk morpholino that can target both maternal and zygotic RNAs (**Figure 4—figure supplement 1I**).

To determine the effect of *mib1* loss of function in embryos that are totally devoid of Ryk activity, we introduced the *mib1^{tfi91}* mutation in the MZ *ryk^{nce4g}* mutant genetic background. MZ *ryk^{nce4g};-mib1^{tfi91}* double mutants display no significant difference in CE compared to MZ *ryk^{nce4g}* single mutants (**Figure 4K**). This observation is in agreement with our model that the genetic inactivation of *mib1* results in a specific impairment of Ryk activity. Accordingly, Mib1 loss of function has no further consequences on CE in animals that are already devoid of Ryk.

## *Mindbomb1* interacts *genetically with Wnt5b to control convergence extension*

Previous work in zebrafish showed that Ryk interacts with the PCP ligand Wnt5b to control gastrulation movements (**Lin et al., 2010**). If Mib1 is required for Ryk-dependent PCP signaling, Mib1 loss of function should be expected to impair Wnt5b/Ryk signaling. To address this issue, a wnt5b morpholino that has been previously used and validated in different studies (**Kilian et al., 2003**; **Lele et al., 2001**) was injected into Mib1-depleted embryos. Our experiments show that the injection of a subliminal dose of Wnt5b morpholino that has no effect in wild-type controls is already sufficient to significantly enhance the CE defects of mib1 morphants (**Figure 5A**).

The existence of functional interactions between *mib1* and *wnt5b* was further confirmed through wnt5b morpholino injections in *mib1^{tfi91}* mutants. Interestingly, the same dose of wnt5b morpholino that we showed to have no effect in ABTÜ wild-type controls (**Figure 5A**) not only decreased CE defects in *mib1^{tfi91}* homozygous mutants (**Figure 5C**), but also had a significant effect on CE in their WT siblings (**Figure 5B**). This later observation suggests that the reduction of *mib1* gene dosage that is already present in these WT sibling embryos is already sufficient to reveal the existence of genetic interactions between *mib*1 and *wnt5b*. In accordance with the previously reported function of Wnt5b/

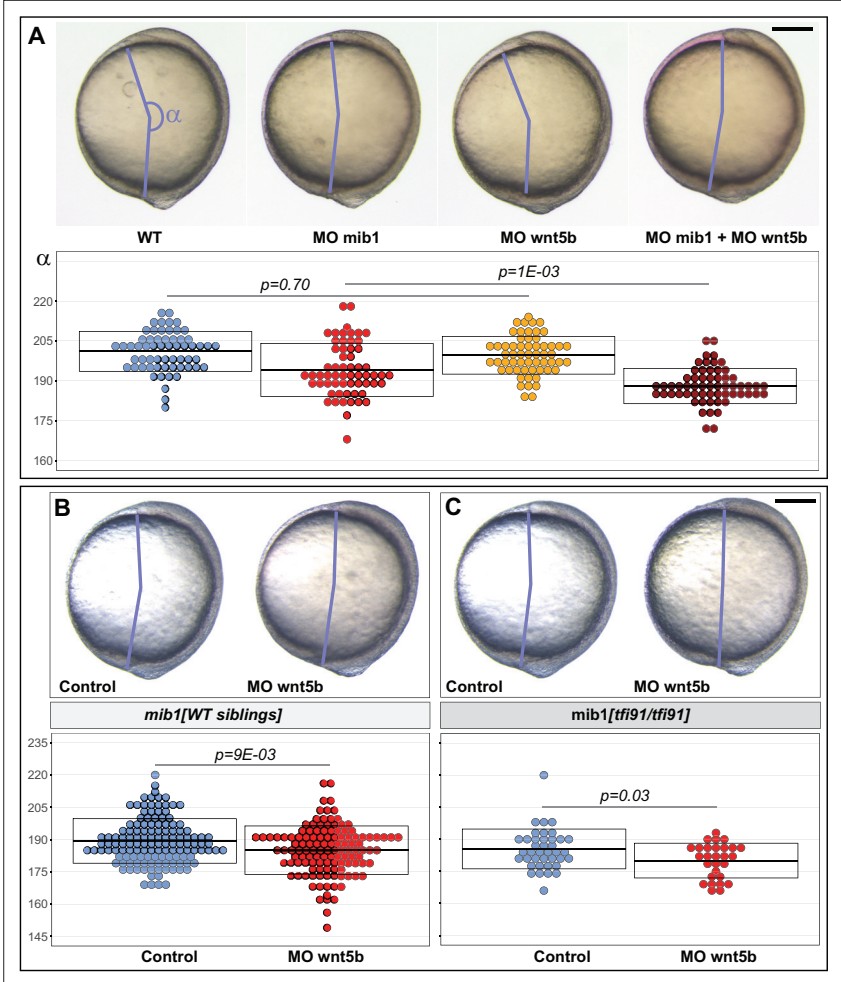

**Figure 5.** *mib1* interacts with *wnt5b* to control Convergence Extension movements. (**A**) Analysis of the axis extension angle α in bud stage embryos reveals that a subliminal dose of wnt5b morpholino (MO wnt5b) that has no effect on Convergent Extension (CE) in WT significantly enhances the defects observed in animals injected with mib1 morpholino (MO mib1). (**B,C**) wnt5b morpholino injection impairs CE in mib1[tfi91] WT siblings (**B**) and enhances CE defects in mib1[tfi91] homozygous mutants (**C**). (**A–C**) Lateral views of bud stage embryos, anterior up, dorsal to the right. Scalebars 200 µm. Boxes represent mean values ± SD. See *Figure 5—source data 1* for complete statistical information.

The online version of this article includes the following source data and figure supplement(s) for figure 5:

**Source data 1.** Complete statistical information for the experiments reported in *Figure 5*.

**Figure supplement 1.** Mib1 is dispensable for Wnt5b-induced Ryk degradation.

Ryk signaling in zebrafish gastrulation (*Lin et al., 2010*), our findings therefore suggest that through its ability to control Ryk localization, Mib1 may act as a modulator of Wnt5b-dependent CE movements.

Mib1 could modulate Wnt5b/Ryk signaling by controlling overall cellular Ryk distribution or promoting a specific, ligand-dependent internalization of Ryk upon Wnt5b stimulation. To address this issue, we compared the impact of Wnt5b or Mib1 overexpression on Ryk localization. Our experiments reveal that overexpression of high amounts of Mib1 (125 pg, *Figure 3—figure supplement 1F*) or Wnt5b (300 pg, *Figure 5—figure supplement 1A, B*) cause an overall reduction of Ryk-GFP levels, in accordance with the previously reported function of Mib1 in Ryk degradation (*Berndt et al., 2011*). While the overexpression of lower amounts of Mib1 results in a clear relocalization of Ryk-GFP from the plasma membrane to intracellular compartments (*Figure 3B*, *Figure 3—figure supplement 1C, E*), we never observed a similar intracellular accumulation of Ryk upon Wnt5b misexpression. To ascertain that failure to observe Wnt5b-mediated intercellular Ryk accumulation was not due to

specific experimental conditions, we repeated these experiments using a range of concentrations of different Wnt5b and Ryk constructs as well as different misexpression and staining protocols. Under no circumstances did we however observe a clear Wnt5b-induced relocalization of Ryk to intracellular compartments similar to the one induced by Mib1 misexpression (*Figure 3—figure supplement 1C, E*). The different effects of Wnt5b and Mib1 overexpression suggest that Mib1 is not involved in the Wnt5b-dependent control of Ryk localization. In accordance with this hypothesis, the loss of Ryk-GFP expression that is observed upon overexpression of high amounts of Wnt5b does not require Mib1 function (*Figure 5—figure supplement 1C, D*).

Taken together our findings identify the E3 Ubiquitin ligase Mib1 as a novel regulator of PCP-dependent CE movements. While the precise cellular and molecular mechanism that underlies Mib1 function in PCP signaling remains to be established, we provide evidence that through its ability to control endocytic Ryk internalization, Mib1 acts as an essential modulator of Ryk/Wnt5b-mediated PCP signaling.

## Discussion

Vertebrate Notch signaling involves multiple ligands, receptors and downstream transcription factors, but the internalization of Delta ligands that triggers Notch receptor activation is regulated essentially by Mib1 (*Koo et al., 2007*; *Mikami et al., 2015*). For this reason, numerous studies have turned to the functional inactivation of *mib1* to inhibit Notch signaling in different biological contexts. A number of recent studies have however shown that the role of Mib1 extends beyond the Notch pathway (*Li et al., 2011*; *Mizoguchi et al., 2017*; *Sturgeon et al., 2016*; *Villumsen et al., 2013*; *Cajanek et al., 2015*). In the present study, we have identified a novel Notch-independent function of Mib1 in the regulation of PCP-dependent CE movements during zebrafish gastrulation.

We show that two potential *mib1* null alleles, *mib1*[tfi91] and the newly generated *mib1*[nce2a] (which retain only the first 59 or 57 amino acids of the 1130 residue wild type protein), cause defects in gastrulation stage axis extension movements (*Figure 1G*). Similar defects are observed in mib1 morphants (*Figure 1A*) but not in *mib1*[ta52b] mutants that present a missense mutation in the C-terminal RF domain (*Figure 1D*). In the context of Delta/Notch signaling, the Mib1[ta52b] mutant protein exerts a dominant-negative activity and thereby causes Notch loss of function phenotypes that are even more severe than the ones observed for *mib1*[tfi91] null mutants (*Zhang et al., 2007b*). The fact that *mib1*[ta52b] mutants present no defects in embryonic CE suggests therefore that the role of Mib1 in CE is distinct from its function in Notch signaling. This hypothesis is further substantiated by the finding that the CE phenotypes of mib1-depleted animals can be rescued using the PCP pathway component RhoA (*Figure 2A*), but not with constitutively active Notch (*Figure 1E*).

All known Mib1 functions require the C-terminal RF domains that are key for the protein's E3 ubiquitin ligase activity. In accordance with a similar mode of action, a Mib1 variant that lacks all three RF domains is unable to support proper CE (*Figure 2B*). While axis extension occurs normally in *mib1*[ta52b] RF3 point mutants, a Mib1 RF3 deletion variant is unable to support CE (*Figure 2C*). These observations suggest that the capacity of Mib1 to mediate substrate ubiquitination is required for the PCP-dependent control of CE movements, but raise the question how the *mib1*[ta52b] mutation may specifically impair Notch but not PCP signaling?

The N-terminal part of Mib1 has been shown to interact with different protein substrates (*Berndt et al., 2011*; *Guo et al., 2016*). Our observations show that the contribution of Mib1 to specific signaling pathways can also be impaired through alterations in RF3. The RF domains of E3 ubiquitin ligases interact with ubiquitin-conjugating E2 enzymes (*Guo et al., 2016*). While the importance of different E3 ligases for specific signaling pathways is well established, it is less clear if different E2 enzymes (of which there are around 40 in the human genome) also contribute to the pathway-specific control of cell signaling. A possible – but currently entirely speculative – explanation for our observation that *mib1*[ta52b] mutants present impaired Notch but intact PCP signaling could be that the *ta52b* RF3 point mutation disrupts the interaction of Mib1 with a specific E2 enzyme required for Notch signaling, while having no effect on a distinct E2 enzyme involved in PCP.

Mib1 promotes Ryk endocytosis in mammalian cell culture (*Berndt et al., 2011*), but the functional relevance of this interaction for vertebrate development has not been addressed. Studies in *Xenopus* and zebrafish identified Ryk functions in PCP-dependent morphogenetic processes (*Kim et al., 2008*;

*Lin et al., 2010*; *Macheda et al., 2012*). Our experiments in *mib1* null mutants and mib1 morphants identify Mib1 as an essential regulator of Ryk endocytosis and Ryk-dependent CE (*Figure 3E–L*).

Both *mib1* mutants and mib1 morphants present a partial impairment of Ryk endocytosis (*Figure 3J*). Due to this situation, we were able to restore the number of Ryk-positive endosomes by Ryk overexpression (*Figure 3E*). The observation that Ryk overexpression not only rescues the number of Ryk-positive endosomes but also the CE movements of mib1-depleted animals (*Figure 3K and L*) suggests that Ryk-positive endosomal compartments are required for PCP signaling.

The partial loss of Ryk-positive endosomes that is observed in *mib1* mutants is most likely due to the perdurance of maternally deposited products. The mib1 splice morpholino used in the present study blocks zygotic mib1 production but has no effect on maternally deposited Mib1 protein. In contrast, the Mib1^{ΔRF123} protein retains the N-terminal Ryk interaction domain (*Berndt et al., 2011*) while lacking the C-terminal RF domains required for substrate ubiquitination. In analogy with studies of Mib1^{ΔRF123} in Delta/Notch signaling (*Itoh et al., 2003*; *Mikami et al., 2015*; *Zhang et al., 2007a*), we expect this Mib1 protein variant to exert a dominant-negative activity by sequestering Ryk without promoting its ubiquitination. Accordingly, the most severe inhibitions of Ryk endocytosis are observed in embryos that have been co-injected with mib1 splice morpholino and dominant-negative Mib1^{ΔRF123}.

As a growing number of Mib1 substrates are being identified (*Matsuda et al., 2016*; *Mertz et al., 2015*; *Mizoguchi et al., 2017*; *Tseng et al., 2014*), the question arises whether Mib1 could regulate the ubiquitin-dependent trafficking of additional PCP pathway components. Our observations do not support this hypothesis: First, Mib1 overexpression specifically promotes Ryk internalization, while having no discernable effect on the localization of other PCP-related transmembrane proteins (*Figure 3A–D*, *Figure 3—figure supplement 3*). Second, our analysis of *mib1;ryk* double mutants shows that the CE defects that are already observed in animals that are entirely devoid of Ryk are not further enhanced by the loss of Mib1 (*Figure 4K*).

Despite the fact that *mib1* null mutants and mib1 morphants present a similar reduction in the number of Ryk-positive endosomes (*Figure 3J*), CE defects are more severe in morphants than mutants (*Figure 1G*). The *mib1^{tfi91}* and *mib1^{nce2a}* alleles used in our study introduce early stop codons in the *mib1* open reading frame, a mutation pattern that has been reported to induce degradation of mutant transcripts and upregulation of compensatory genes (*El-Brolosy et al., 2019*). In accordance with such a scenario, *mib1* transcript levels are reduced in these *mib1* mutants (*Figure 1I*, *Figure 1—figure supplement 2D, E*). While compensation could potentially occur through a mechanism that promotes Mib1-independent Ryk endocytosis, our data do not support such a hypothesis (*Figure 3J*). Instead, our observations indicate a mechanism of PCP pathway resilience that allows to partially correct the defects that arise from a failure in Ryk endocytosis.

Ryk has been shown to interact with the PCP ligand Wnt5b to regulate zebrafish gastrulation (*Lin et al., 2010*). While the precise molecular mechanism through which Mib1 controls PCP-dependent CE movements remains to be established, our observation of genetic interactions between *mib1* and *wnt5b* (*Figure 5*) suggests that Mib1 acts as a modulator of Ryk/Wnt5b-mediated PCP signaling.

A major open question for future studies will be to determine whether Mib1 is also required for the control of PCP-dependent processes in other biological contexts. While an analysis of mib1 function in zebrafish spinal cord morphogenesis revealed several phenotypes that could be potentially linked to PCP signaling (*Sharma et al., 2019*; VMS, MF, unpublished observations), it remains to be established whether these defects are indeed due to impaired PCP activity, or arise from the implication of Mib1 in Notch signaling (*Itoh et al., 2003*). In this context, the analysis of gastrulation stage CE movements, which are critically dependent on PCP signaling but do not require Notch, provided a unique opportunity to unambiguously establish a function of Mib1 in the control of PCP-dependent morphogenetic movements.

Taken together, our findings identify the E3 ubiquitin ligase Mib1 as an essential novel regulator of PCP-dependent CE movements during zebrafish gastrulation. Mib1 regulates CE movements through the control of Ryk endocytosis, independent of its role in Delta/Notch signaling. As processes such

as the morphogenesis of the vertebrate neural tube involve both Notch and PCP signaling, our data suggest that great care should be taken when interpreting Mib1 loss-of-function phenotypes.

## Materials and methods

**Key resources table**

| Reagent type (species) or resource | Designation | Source or reference | Identifiers | Additional information |
|---|---|---|---|---|
| Gene (*Danio rerio*) | *mib1* | | ZDB-GENE-030404–2 | |
| Gene (*Danio rerio*) | *Ryk* | | ZDB-GENE-070209–277 | |
| Genetic reagent (*Danio rerio*) | *mib1^ta52b^* | DOI:10.1016/s1534-5807(02)00409-4 | ZDB-ALT-980203–1374 | |
| Genetic reagent (*Danio rerio*) | *mib1^tfi91^* | DOI:10.1016/s1534-5807(02)00409-4 | ZDB-ALT-060208–4 | |
| Genetic reagent (*Danio rerio*) | *mib1^nce2a^* | This paper | | See material and methods |
| Genetic reagent (*Danio rerio*) | *ryk^nce4g^* | This paper | | See material and methods |
| Sequence-based reagent | Crispr-mib1-166 | This paper | | See material and methods |
| Sequence-based reagent | Crispr-ryk-59251 | This paper | | See material and methods |
| Sequence-based reagent | mib1 exon/ intron one splice morpholino | DOI:10.1016/s1534-5807(02)00409-4 | | See material and methods |
| Sequence-based reagent | ryk morpholino | This paper | | See material and methods |
| Sequence-based reagent | wnt5b morpholino | DOI:10.1002/gene.1063. | | See material and methods |
| Recombinant DNA reagent | Mib1^ta52b^-pCS2+ | DOI:10.1016/j.jmb.2006.11.096 | | |
| Recombinant DNA reagent | Mib1^ΔRF123^-pCS2+ | DOI:10.1016/j.jmb.2006.11.096 | | |
| Recombinant DNA reagent | Mib1-pCS2+ | This paper | | See material and methods |
| Recombinant DNA reagent | Mib1-ΔRF3-pCS2+ | This paper | | See material and methods |
| Recombinant DNA reagent | Flag-Ryk-Myc-pCS2+ | DOI:10.1083/jcb.200912128 | | |
| Recombinant DNA reagent | Ryk-GFP-pCS2+ | DOI:10.1083/jcb.200912128 | | |
| Recombinant DNA reagent | Ryk-pCS2+ | This paper | | See material and methods |
| Recombinant DNA reagent | Ryk^nce4g^-pCS2+ | This paper | | See materials and methods |
| Recombinant DNA reagent | Ryk-HA-pCS2+ | This paper | | See materials and methods |
| Recombinant DNA reagent | Ryk^nce4g^-HA-pCS2+ | This paper | | See materials and methods |
| Recombinant DNA reagent | Ryk-GFP-pCS2+ | This paper | | See materials and methods |
| Recombinant DNA reagent | GFP-Vangl2-pCS2+ | DOI:10.1083/jcb.201111009 | | |
| Recombinant DNA reagent | Fz2-mCherry-pCS2+ | DOI:10.1083/jcb.200912128 | | |
| Recombinant DNA reagent | Fz7-YFP-pCS2+ | DOI:10.1083/jcb.200606017 | | |
| Recombinant DNA reagent | ROR1-GFP-pCS2+ | This paper | | See materials and methods |

*Continued on next page*

*Continued*

| Reagent type (species) or resource | Designation | Source or reference | Identifiers | Additional information |
|---|---|---|---|---|
| Recombinant DNA reagent | ROR2-mCherry-pCS2+ | DOI:10.7554/eLife.36953 | | |
| Recombinant DNA reagent | GFP-Rab5c-pCS2+ | This paper | | See materials and methods |
| Recombinant DNA reagent | RhoA-pCS2+ | DOI:10.1038/ncb2632 | | |
| Recombinant DNA reagent | NICD-pCS2+ | PMID:10357943 | | |
| Recombinant DNA reagent | Histone2B-mRFP-pCS2+ | DOI:10.1038/nature02796 | | |
| Recombinant DNA reagent | Histone2B-GFP-pCS2+ | This paper | | See materials and methods |
| Recombinant DNA reagent | Histone2B-tagBFP-pCS2+ | This paper | | See materials and methods |
| Recombinant DNA reagent | Wnt5b-pCS2+ | DOI:10.1083/jcb.200912128 | | |
| Sequence-based reagent | mib1 qPCR primer (fwd) | This paper | | See materials and methods |
| Sequence-based reagent | mib1 qPCR primer (rev) | This paper | | See materials and methods |
| Sequence-based reagent | ryk qPCR primer (fwd) | This paper | | See materials and methods |
| Sequence-based reagent | ryk qPCR primer (rev) | This paper | | See materials and methods |
| Sequence-based reagent | 36b4 qPCR primer (fwd) | This paper | | See materials and methods |
| Sequence-based reagent | 36b4 qPCR primer (rev) | This paper | | See materials and methods |
| Antibody | Anti-HA 3F10 (rat monoclonal) | Roche | # 11867423001, RRID:AB_390918 | (1:500) |
| Antibody | Anti-cMyc 9E10 (mouse monoclonal) | Santa Cruz | # sc-40, RRID:AB_2857941 | (1:500) |
| Software, algorithm | ImageJ/Fiji | https://imagej.net/software/fiji/ | | |
| Software, algorithm | R | https://www.r-project.org/ | | |
| Software, algorithm | RStudio | https://www.rstudio.com/ | | |

## Crispr/Cas mutagenesis

### Generation of *mib1*[nce2a] mutants

Crispr/Cas9 mutagenesis of zebrafish *mib1* was performed using the reverse strand exon 1 target site 5'- GGAGCAGCGGTAATTGGCGGCGG-3' (bold lettering indicates the PAM motif). gRNA design and in vitro transcription were performed according to reported protocols (*Hruscha et al., 2013*; *Jao et al., 2013*). A pre-assembled complex of purified Cas9 protein (NEB) and gRNA was injected and the efficiency of Crispr/Cas9-induced mutagenesis in the F0 generation monitored at 24 hpf using a T7 endonuclease assay (*Jao et al., 2013*) on a PCR amplicon comprising the Crispr target region (Forward primer: 5'- TGACTGGAAGTGGGGGAAGC-3', Reverse primer: 5'- TGCAGTATTAGAAACGCGTG-3'). Direct sequencing of the same PCR amplicon was used to identify induced mutations in the F1 generation. This procedure led to the identification of the *mib1*[nce2a] mutant allele which introduces a frame shift after amino acid 57 and induces the appearance of a premature stop codon after residue 69.

## Generation of *ryk*^nce4g mutants

Crispr/Cas9 mutagenesis of zebrafish *ryk* was performed using the reverse strand exon 5 target site 5'- GGCAGAGTTTTGGGGGGCTCTGG-3' using the strategy mentioned above. The T7 endonuclease assay in the F0 generation and mutation identification in F1 were performed using the same PCR amplicon (Forward primer: 5'-GTGATGTTAGACTTGCATAC-3', Reverse primer: 5'-GAAGGTTTACAA GGGCAGAATG-3'). The *ryk*^nce4g mutation introduces an 11 bp insertion that causes a frame shift after amino acid 196 and induces the appearance of a premature stop codon after residue 214.

## Fish strains and molecular genotyping

Unless otherwise specified, experiments were performed in embryos derived from an ABTÜ hybrid wild-type strain. Mutant strains included *mib1*^ta52b (**Itoh et al., 2003**), *mib1*^tfi91 (**Itoh et al., 2003**), *mib1*^nce2a (this study) and *ryk*^nce4g (this study).

Depending on the experiment, *mib1* homozygous mutants were identified using molecular geno-typing (see below) or through the identification of the characteristic white tail phenotype that can easily be identified by 36 hpf.

For the genotyping of *mib1*^ta52b mutants a 4-primer-PCR was used to identify WT and mutant alleles in a single PCR. The primers used were: 5'-ACAGTAACTAAGGAGGGC-3' (generic forward primer), 5'-AGATCGGGCACTCGCTCA-3' (specific WT reverse primer), 5'-TCAGCTGTGTGGAGACCGCAG-3' (specific forward primer for the *mib1*^ta52b allele), and 5'-CTTCACCATGCTCTACAC-3' (generic reverse primer). WT and *mib1*^ta52b mutant alleles respectively yield 303 bp and 402 bp amplification fragments. As some zebrafish strains present polymorphic *mib1* WT alleles, it is important to validate the applica-bility of this protocol before use in a given genetic background.

For the genotyping of *mib1*^tfi91 mutants, two separate allele-specific PCRs were used to identify WT and mutant alleles. The primers used were: 5'-TAACGGCACCGCCGCCAATTAC-3' and 5'- GCGA CCCCAGATTAATAAAGGG-3' (WT allele), 5'-ATGACCACCGGCAGGAATAACC-3' and 5'- ACATCATA AGCCCCGGAGCAGCGC-3' (mutant allele).

For the genotyping of *mib1*^nce2a mutants, two separate allele-specific PCRs were used to identify WT and mutant alleles. The primers used were: 5'-GCAGGAATAACCGAGTGATG-3' and 5'- AGCA GCGGTAATTGGCGG-3' (WT allele), 5'-GCAGGAATAACCGAGTGATG-3' and 5'- GAGCAGCGGTAA TTGAATA-3' (mutant allele).

For the genotyping of *ryk*^nce4g mutants, a single PCR reaction was used to amplify the mutation-carrying region (Forward primer 5'-GTGATGTTAGACTTGCATAC-3', Reverse primer 5'-GAAGGTTT ACAAGGGCAGAATG-3'). Due to the presence of an 11 bp insertion, mutant and WT alleles can be distinguished on a 2.5% agarose gel.

To avoid issues related to variations among genetic backgrounds, the different adult fish used in the course of our analysis of *ryk*^nce4g single and *ryk*^nce4g; *mib1*^tfi91 double mutants (**Figure 4G, H**, **Figure 4—figure supplement 1C, E, F**) were all derived from a single incross of *ryk*^nce4g/+; *mib1*^tfi91/+ parents.

For DNA extraction embryos were lysed 20 min at 95 °C in 28.5 µl 50 mM NaOH, and then neutral-ized by adding 1.5 µl Tris-HCl pH 7.5. PCR amplifications were carried out using GoTaq G2 polymerase (Promega) at 1.5 mM MgCl2 using the following cycling parameters: 2 min 95 °C - 10 cycles [30 sec 95 °C – 30 sec 65°C to 55°C – 60 sec 72 °C] – 25 cycles [30 sec 95 °C – 30 sec 55 °C – 60 sec 72 °C] – 5 min 72 °C.

## mRNA and Morpholino injections

Microinjections into dechorionated embryos were carried out using a pressure microinjector (Eppen-dorf FemtoJet). Capped mRNAs were synthesized using the SP6 mMessage mMachine kit (Ambion). RNA and morpholinos were injected together with 0.2% Phenol Red.

Morpholinos were injected at 500 (mib1 5'-GCAGCCTCACCTGTAGGCGCACTGT-3', **Itoh et al., 2003**), 62.5 (ryk 5'-GGCAGAAACATCACAGCCCACCGTC-3') or 250 µM (wnt5b 5'- GTCCTTGGTTCA TTCTCACATCCAT-3', **Lele et al., 2001**).

RNA microinjection was performed using the following constructs and quantities: Mib1^ta52b-pCS2+ (12.5–25 pg) and Mib1^ΔRF123 (125 pg) (**Zhang et al., 2007b**). Mib1-pCS2+ (12.5 pg unless otherwise indicated) and Mib^ΔRF3-pCS2+ (125 pg) (this study). Ryk-GFP-pCS2+ (25 pg) and Flag-Myc-Ryk-pCS2+ (50 pg) (**Lin et al., 2010**). Ryk-pCS2+ (0.75–25 pg), Ryk^nce4g-pCS2+ (1.5–25 pg), Ryk-HA-pCS2+ (25 pg),

Ryk<sup>nce4g</sup>-HA-pCS2+ (25 pg), Ryk-GFP-pCS2+ (3–12 pg) (this study, all constructs have been engineered to abolish ryk morpholino binding without altering the Ryk protein sequence). GFP-Vangl2-pCS2+ (*Mahuzier et al., 2012*). Fz2-mCherry-pCS2+ (50 pg) (*Lin et al., 2010*). Fz7-YFP-pCS2+ (25 pg) (*Witzel et al., 2006*). ROR1-GFP-pCS2+ (25 pg, this study). ROR2-mCherry-pCS2+ (25 pg) (*Mattes et al., 2018*). GFP-Rab5c-pCS2+ (50 pg, this study). RhoA-pCS2+ (25 pg) (*Castanon et al., 2013*). NICD-pCS2+ (37.5 pg) (*Takke and Campos-Ortega, 1999*). GAP43-RFP-pCS2+ (25 pg). Histone2B-mRFP-pCS2+ (12.5 pg) (*Gong et al., 2004*). Histone2B-GFP (6 pg) and Histone2B-tagBFP (1.5 pg) (this study). Wnt5b (300 pg) (*Lin et al., 2010*).

## RNA in situ hybridization

Whole mount RNA in situ hybridizations were performed as previously described (*Thisse and Thisse, 2008*). The *dlx3* probe has been previously described (*Kilian et al., 2003*). *ryk* antisense RNA was transcribed from ryk-pBSK (this study). For *papc, mib1, shhb* and *foxa3*, in situ probes were transcribed from PCR products that contained a T7 promoter sequence at their 3'end. The papc region amplified from genomic DNA extended from 5'-TCCTTCTGCAGCTCGTCCGACTGGAAG-3' (forward strand) to 5'-GGTAAACCACCCACAGTTGAC-3' (reverse). The mib1 probe region amplified from Mib1-pCS2+ extended from 5'-CCCGAGTGCCATGCGTGTGCTGC-3' (forward) to 5'-CGCCGAATCCTGCTTT AC-3' (reverse). *shhb* was amplified using primers 5'-TGTAAAACGACGGCCAGT-3' (forward) and 5'-CAGGAAACAGCTATGACC-3' (reverse). *foxa3* was amplified using 5'-TGTTTTGGGGAAGCAG GAGTCA-3' (forward) and 5'-CGTAATACGACTCACTATAGGGAGATCAGTGAAGAACAGAGAGG TCACT-3' (reverse). shhb-pBSK and foxa3-pSPORT3 were provided by the Thisse lab.

## qPCR analysis

For each biological replicate total RNA was isolated from 50 embryos using TRI-Reagent (Sigma). Reverse transcription was performed on 2.5 µg of RNA using Superscript III (Invitrogen) to generate cDNA. qPCR was performed using PowerUP SYBR Green Master Mix (Applied Biosystems) in an Applied Biosystems Step-One PCR system.

The following primers were used for the amplification of different transcripts:

*mib1* forward: 5'- GTGGTGGTGGTGTGGGATA-3'
*mib1* reverse: 5'- TCGTAGTTGGTGCACTCTGC-3'
*ryk* forward: 5'- GAACATCTTCATGAGCGAGG-3'
*ryk* reverse: 5'- TAGTGTCACTGGGCACTGGTAG-3'

*36b4* forward: 5'- ACGTGGAAGTCCAACTACT-3'
*36b4* reverse: 5'- GTCAGATCCTCCTTGGTGA-3'

Ct values at 40 cycles of qPCR amplification were used for estimating relative gene expression using ΔΔCT method described by *Livak and Schmittgen, 2001*. Fold changes in gene expression were normalized to the internal control gene *36b4*. For each experiment three biological replicates were used. Measurements for each replicate were performed using technical triplicates.

## Immunocytochemistry

Dechorionated embryos were fixed overnight at 4 °C in PEM (80 mM Sodium-Pipes, 5 mM EGTA, 1 mM MgCl$_2$) - 4% PFA - 0.04% TritonX100. After washing 2 × 5 min in PEMT (PEM - 0.2% TritonX100), 10 min in PEM - 50 mM NH4Cl, 2 × 5 min in PEMT and blocking in PEMT - 5% NGS, embryos were incubated 2 hrs at room temperature with primary antibodies. Following incubation, embryos were washed during 5, 10, 15 and 20 min in PEMT, blocked in PEMT - 5% NGS, and incubated again with secondary antibodies for 2 hrs. Embryos were again washed during 5, 10, 15, and 20 min in PEMT. The following primary antibodies were used: Rat@HA (Roche 11 867 423 001, 1:500). Mouse@c-Myc9E10 (Santa Cruz sc-40, 1:500). Secondary antibodies Goat@Rat-Alexa488 (Invitrogen) and Goat@Mouse-Cy5 (Jackson Immunoresearch) were used at a dilution of 1:500.

## Microscopy and image analysis

For confocal imaging, embryos were mounted in 0.75% low melting agarose (Sigma) in glass bottom dishes (Mattek). Embryos were imaged on Spinning disk (Andor) or Laser scanning confocal microscopes (Zeiss LSM710, 780 and 880) using 40 x Water or 60 x Oil immersion objectives. The localisation

of Ryk and other PCP pathway components were analyzed at 90% epiboly stage in the dorsal epiblast. Bud stage axis extension and in situ gene expression patterns were documented on Leica M205FA-Fluocombi or Leica MZ-FLIII stereomicroscopes coupled to Lumenera color CCDs. Image analysis was performed using ImageJ (http://rbs.info.nih.gov/ij/). Quantifications were performed blindfolded without knowledge of the sample genotype.

## Statistical analysis

Statistical analysis was performed using R. Data normality and variance were analyzed using Shapiro-Wilk and Levene's tests and statistical tests chosen accordingly. Complete informations about sample sizes, numerical values, and tests statistics for all experiments are provided in the Source data files.

## Use of research Animals

Animal experiments were performed in the iBV Zebrafish facility (authorization #B-06-088-17) in accordance with the guidelines of the ethics committee Ciepal Azur and the iBV animal welfare committee (project authorizations NCE/2013–92, 19944–2019031818528380).

## Acknowledgements

Confocal microscopy was performed with the help of the iBV PRISM imaging platform. We thank L Bally-Cuif, I Castanon, M Gonzalez-Gaitan, S Guo, CP Heisenberg, M Itoh, B Link, S Scholpp, D Slusarski, B & C Thisse, C Vesque and YJ Jiang for the sharing of fish lines and reagents. We are grateful to R Rebillard for excellent fish care and to T Govekar for help with preliminary experiments. We would like to dedicate this paper to the memory of Bernard Thisse who sadly passed away while this study was under revision.

## Additional information

### Funding

| Funder | Grant reference number | Author |
|---|---|---|
| Fondation ARC pour la Recherche sur le Cancer | PJA20181208167 | Maximilian Fürthauer |
| Agence Nationale de la Recherche | ANR-17-CE13-0024-02 | Maximilian Fürthauer |
| Agence Nationale de la Recherche | ANR-11-LABX-0028-01 | Vishnu Muraleedharan Saraswathy |
| Ligue Contre le Cancer | IP/SC-17131 | Akshai Janardhana Kurup |
| Fondation pour la Recherche Médicale | FDT20140930987 | Priyanka Sharma |
| Human Frontier Science Program | CDA00036/2010 | Maximilian Fürthauer |
| Centre National de la Recherche Scientifique | ATIP2010 | Maximilian Fürthauer |
| Fondation ARC pour la Recherche sur le Cancer | A2011 Postdoctoral fellowship | Morgane Poulain |

The funders had no role in study design, data collection and interpretation, or the decision to submit the work for publication.

### Author contributions

Vishnu Muraleedharan Saraswathy, Conceptualization, Formal analysis, Investigation, Methodology, Validation, Visualization, Writing - original draft, Writing - review and editing; Akshai Janardhana Kurup, Formal analysis, Funding acquisition, Investigation, Methodology, Validation, Visualization, Writing - original draft, Writing - review and editing; Priyanka Sharma, Conceptualization, Formal

analysis, Funding acquisition, Investigation, Methodology, Validation, Writing - original draft; Sophie Polès, Investigation, Resources; Morgane Poulain, Resources; Maximilian Fürthauer, Conceptualization, Formal analysis, Funding acquisition, Investigation, Methodology, Project administration, Supervision, Validation, Visualization, Writing - original draft, Writing - review and editing

**Author ORCIDs**
Maximilian Fürthauer ⓘ http://orcid.org/0000-0001-6344-6585

**Ethics**
Animal experiments were performed in the iBV Zebrafish facility (authorization #B-06-088-17) in accordance with the guidelines of the ethics committee Ciepal Azur and the iBV animal welfare committee (project authorizations NCE/2013-92, 19944-2019031818528380).

**Decision letter and Author response**
Decision letter https://doi.org/10.7554/eLife.71928.sa1
Author response https://doi.org/10.7554/eLife.71928.sa2

## Additional files

### Supplementary files
• Transparent reporting form

### Data availability
All data generated or analysed during this study are included in the manuscript.

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
