## [Decision Letter]

**Decision letter after peer review:**

Thank you for submitting your article "The E3 Ubiquitin Ligase Mindbomb1 controls zebrafish Planar Cell Polarity" for consideration by *eLife*. Your article has been reviewed by 3 peer reviewers, including Ajay Chitnis as the Reviewing Editor and Reviewer #1, and the evaluation has been overseen by Didier Stainier as the Senior Editor.

The authors demonstrate that impaired *mib1* function is associated with both increased accumulation of Ryk-GFP on the cell surface and reduced intracellular Ryk-GFP in an endocytic compartment. Furthermore, they show and that convergence extension (CE) defects in embryos with impaired Mib1 function is likely to be associated with the reduced Ryk endocytosis. The authors go on to demonstrate that effects of Mib1 on internalization of Ryk are specific and not seen with other PCP components like Vangl2, Fz2 or Fz7. It is noteworthy that Mib over-expression does not affect Vangl2 localization, as previous studies have identified a genetic interaction between Ryk and Vangl2 (Macheda et al. 2012).

Finally, as Mib1 function is hypothesized to be relevant only in the context of Ryk endocytosis for its CE defects, the authors show that loss of Mib1 function does not increase the severity of these defects in Ryk null mutants.

The strength of the manuscript is that Mib1 regulates PCP-dependent CE in a Notch-independent manner. Also, the genetic epistasis analysis demonstrates that Mib1 loss of function has no effect on CE in maternal zygotic ryk null mutants, suggesting that Mib1 is an essential regulator of Ryk endocytosis during zebrafish CE. These demonstrations are convincing. However, the weakness is the conclusion that Mib1 is an 'essential' regulator of PCP is not fully supported by the data. Rather, Mib1 could be a context-dependent modulator of PCP. Clarification of this weakness would make conclusions of this paper significantly stronger.

It also remains unclear how Ryk endocytosis contributes to effective PCP or how reduced Ryk endocytosis alters the organization or function of other PCP components.

This study highlights the need to consider complex signaling networks when evaluating phenotypes, especially convergent extension.

Essential revisions:

1. Provide a clarification of why the *mib1* ta52b allele does not interfere with Mib1 function in the context of Ryk endocytosis. Can Mib ta52bb promote Ryk endocytosis?

2. The title 'The E3 Ubiquitin Ligase Mindbomb1 controls zebrafish Planar Cell Polarity' is misleading. There is no evidence that Mib1 regulates PCP in processes other than CE, independently of Notch signaling. Strictly speaking, Mib1 regulates endocytosis of Ryk, which modulates PCP during CE. Considering the fact that Mib1 also modulates Wnt/b-catenin signaling through Ryk (Berndt, 2011), it is too strong to state in the abstract that 'We show that Mib1, a known Notch signaling regulator, is also an essential PCP pathway component'. It is likely that Ryk is a context-dependent modulator of PCP rather than the PCP component, for example, by modulating Vangl2 (Macheda, 2012).

Since maternal Ryk modulates PCP-dependent CE, one possible scenario is that the Wnt5b-Ryk axis regulates CE, which is dependent on Mib1. Several lines of evidence supports this; (i) MZwnt5b/ppt embryos show CE defects (Kilian, 2005), (ii) Wnt5b mediates Ryk endocytosis (Lin, 2010).

Our suggestion is to examine (i) whether Mib1 genetically interacts with wnt5b/ppt (using MO) and (ii) Wnt5 induced Ryk endocytosis is suppressed in *mib1* null embryos expressing a dominant negative Mib1 (Mib^ΔRF123^).

*Reviewer #1 (Recommendations for the authors):*

The paper describes a straightforward set of logical experiments to explore the link between Mib1, Ryk and its role in CE. The primary conclusions of the paper are supported by data presented. Nevertheless, many questions remain unanswered. It remains unclear to me why the ta52b allele has such a minimal effect, the authors could elaborate on this. It also remains unclear how Ryk endocytosis contributes to effective PCP or how reduced Ryk endocytosis alters the organization or function of other PCP components. However, it is reasonable for these questions to be addressed in future studies.

The authors have provided information about relevant data, reagents pertaining to *eLife* policy.

*Reviewer #2 (Recommendations for the authors):*

The authors make the conclusion that the different phenotypes between ta52b and the proposed null alleles support a novel role for mib in regulating CE. While the authors expand on in the discussion, it is unclear to this reviewer if the ta52b is notch-specific or a hypo-morphic mutation. The ΔRF123 is used as a dominant negative for Mib function. Better characterization of this reagent would be useful.

A comment in the manuscript that needs to be reconciled. The author state how the mib morphants have the most severed CE compared to the putative null alleles, yet they use the anti-morphic Δ-ring construct to further deplete mib function. This is of concern to this reviewer.

*Reviewer #3 (Recommendations for the authors):*

The title 'The E3 Ubiquitin Ligase Mindbomb1 controls zebrafish Planar Cell Polarity' is misleading. There is no evidence that Mib1 regulates PCP in processes other than CE, independently of Notch signaling. Strictly speaking, Mib1 regulates endocytosis of Ryk, which modulates PCP during CE. Considering the fact that Mib1 also modulates Wnt/b-catenin signaling through Ryk (Berndt, 2011), it is too strong to state in the abstract that 'We show that Mib1, a known Notch signaling regulator, is also an essential PCP pathway component'. It is likely that Ryk is a context-dependent modulator of PCP rather than the PCP component, for example, by modulating Vangl2 (Macheda, 2012).

Since maternal Ryk modulates PCP-dependent CE, one possible scenario is that the Wnt5b-Ryk axis regulates CE, which is dependent on Mib1. Several lines of evidence supports this; (i) MZwnt5b/ppt embryos show CE defects (Kilian, 2005), (ii) Wnt5b mediates Ryk endocytosis (Lin, 2010).

My suggestion is to examine (i) whether Mib1 genetically interacts with wnt5b/ppt (using MO) and (ii) Wnt5 induced Ryk endocytosis is suppressed in *mib1* null embryos expressing a dominant negative Mib1 (Mib^ΔRF123^).

---

## [Author Response]

Essential revisions:1. Provide a clarification of why the mib1 ta52b allele does not interfere with Mib1 function in the context of Ryk endocytosis. Can Mib ta52bb promote Ryk endocytosis?

In the revised version of our manuscript we now provide experimental evidence that, like wild-type Mib1, Mib1^ta52b^ is indeed still able to promote Ryk endocytosis The corresponding data are displayed in the newly included Figure 3—figure supplement 2 and mentioned in the Results section:

“If Mib1-dependent Ryk endocytosis is important for CE, this process should be unaffected by the *mib1^ta52b^* mutation that disrupts Notch signaling (Itoh et al., 2003) but has no effect on gastrulation movements (Figure 1D). In accordance with this hypothesis, Mib^ta52b^ misexpression promotes a relocalization of Ryk from the plasma membrane towards intracellular compartments (Figure 3—figure supplement 2A,B), similar to the effect observed upon overexpression of wild-type Mib1 (Figure 3—figure supplement 1B,C).”

While these new experimental data provide a mechanistic explanation for the observations that 1, *mib1^ta52b^* mutants do not present CE defects (Figure 1D) and that 2, *mib1^ta52b^* RNA injection allows to rescue *mib1* morphant CE defects (Figure 1F), it remains to be established how the ta52b mutation specifically disrupts Notch but not PCP signaling. In the Discussion section we provide a potential explanation for this observation, but it is entirely clear that addressing this issue will require further studies that extend beyond the scope of the present work:

“These observations suggest that the capacity of Mib1 to mediate substrate ubiquitination is required for the PCP-dependent control of CE movements, but raise the question how the *mib1^ta52b^* mutation may specifically impair Notch but not PCP signaling? […] A possible – but currently entirely speculative – explanation for our observation that *mib1^ta52b^* mutants present impaired Notch but intact PCP signaling could be that the ta52b RF3 point mutation disrupts the interaction of Mib1 with a specific E2 enzyme required for Notch signaling, while having no effect on a distinct E2 enzyme involved in PCP.”

2. The title 'The E3 Ubiquitin Ligase Mindbomb1 controls zebrafish Planar Cell Polarity' is misleading. There is no evidence that Mib1 regulates PCP in processes other than CE, independently of Notch signaling. Strictly speaking, Mib1 regulates endocytosis of Ryk, which modulates PCP during CE. Considering the fact that Mib1 also modulates Wnt/b-catenin signaling through Ryk (Berndt, 2011), it is too strong to state in the abstract that 'We show that Mib1, a known Notch signaling regulator, is also an essential PCP pathway component'. It is likely that Ryk is a context-dependent modulator of PCP rather than the PCP component, for example, by modulating Vangl2 (Macheda, 2012).

We agree that our current study provides evidence for an implication of Mib1 in the PCP-dependent regulation of gastrulation stage convergence extension movements but does not allow to establish Mib1 as a general PCP regulator. To avoid overinterpretation of our data, we have changed the title of our manuscript to “The E3 Ubiquitin Ligase Mindbomb1 controls PCP-dependent Convergent Extension movements during zebrafish gastrulation”.

Throughout the manuscript we have rephrased the text to clarify that our study establishes Mib1 as a regulator of PCP-dependent CE movements. In the Discussion section, we moreover specifically acknowledge that an important objective for future work will be to determine if Mib1 is involved for PCP signaling in other biological contexts:

“A major open question for future studies will be to determine whether Mib1 is also required for the control of PCP-dependent processes in other biological contexts. […] In this context, the analysis of gastrulation stage CE movements, which are critically dependent on PCP signaling but do not require Notch, provided a unique opportunity to unambiguously establish a function of Mib1 in the control of PCP-dependent morphogenetic movements. ”

Since maternal Ryk modulates PCP-dependent CE, one possible scenario is that the Wnt5b-Ryk axis regulates CE, which is dependent on Mib1. Several lines of evidence supports this; (i) MZwnt5b/ppt embryos show CE defects (Kilian, 2005), (ii) Wnt5b mediates Ryk endocytosis (Lin, 2010).Our suggestion is to examine (i) whether Mib1 genetically interacts with wnt5b/ppt (using MO) and (ii) Wnt5 induced Ryk endocytosis is suppressed in mib1 null embryos expressing a dominant negative Mib1 (Mib^ΔRF123^).

In the revised version of our manuscript, we have now introduced a number of new experiments to address the functional relationship between Mib1 and Wnt5b/Ryk signaling.

In a first series of experiments, we introduced a previously validated wnt5b morpholino into Mib1 depleted (*mib1* morphant or *mib1^tfi91^* mutant) animals. In accordance with a functional relationship between Mib1 and Wnt5b-mediated PCP signaling, we observed that a dose of wnt5b morpholino that has no effect in WT controls is sufficient to generate and/or enhance CE defects in Mib1-depleted animals. The corresponding experiments are displayed in the newly included Figure 5 and presented in the Results section:

“Previous work in zebrafish showed that Ryk interacts with the PCP ligand Wnt5b to control gastrulation movements (Lin et al., 2010). […] In accordance with the previously reported function of Wnt5b/Ryk signaling in zebrafish gastrulation (Lin et al., 2010), our findings therefore suggest that through its ability to control Ryk localization, Mib1 may act as a modulator of Wnt5b-dependent CE movements.”

In a second series of experiments, we attempted to determine if Mib1 would be required for Wnt5b-mediated Ryk internalization. Unexpectedly, these experiments were however hampered by the fact that, in our hands and despite repeated prolonged efforts and the use of a large number of different experimental conditions, the misexpression of Wnt5b failed to promote a relocalization of Ryk from the plasma membrane to intracellular compartments similar to the one observed upon Mib1 overexpression.

To exclude that our failure to detect Wnt5b-driven Ryk accumulation in endocytic compartment was due to specific experimental conditions, we repeated these experiments using a wide range of concentrations of different Wnt5b (Wnt5b, Wnt5b-Myc, Wnt5b-IRES-CAAXGFP, Wnt5b-IRES-NLS-GFP) and Ryk (Ryk-GFP, Ryk-Myc, Ryk-HA, Flag-Ryk-Myc) constructs that were either obtained from previous studies or generated in the course of the present work. Neither the use of these different constructs, nor the use of different misexpression protocols (ubiquitous or localized Wnt5b/Ryk coexpression, Wnt5b and Ryk expression in adjacent cells, transplantation of Ryk-expressing cells into Wnt5b-expressing hosts), nor the use of different antibody staining protocols (PIPES or PBS-based fixation/staining buffers) allowed to provide convincing evidence for Wnt5b-mediated Ryk internalization. While we very occasionally (i.e. few cells of one or two embryos) observed an increase in intracellular Ryk accumulation, we simply never observed a clear relocalization of Ryk from the cell membrane towards intracellular compartments comparable to the one observed upon Mib1 overexpression.

Only at very high doses of Wnt5b RNA (300 pg) we observed a highly penetrant overall decrease of Ryk-GFP levels (Figure 5—figure supplement 1A,B), a phenotype that can also be observed upon overexpression of high amounts of Mib1 (Figure 3—figure supplement 1F). To determine whether the reduction of Ryk levels that is observed upon misexpression of high amounts of Wnt5b is dependent on Mib1 function, we repeated this experiment in *mib1^tfi91^* mutants that were additionally injected with RNA encoding the dominant-negative Mib1 variant Mib1∆RF123. Mib1 loss of function did not impair the Wnt5b-mediated downregulation of Ryk-GFP levels (Figure 5—figure supplement 1C,D), suggesting thereby that Mib1 is not important for the Wnt5b-mediated control of Ryk distribution.

The corresponding data have been integrated in the newly added Figure 5—figure supplement 1 and mentioned in the Results section:

“Mib1 could modulate Wnt5b/Ryk signaling by controlling overall cellular Ryk distribution or promoting a specific, ligand-dependent internalization of Ryk upon Wnt5b stimulation. […] In accordance with this hypothesis, the loss of Ryk-GFP expression that is observed upon overexpression of high amounts of Wnt5b does not require Mib1 function (Figure 5—figure supplement 1C,D).”

Reviewer #1 (Recommendations for the authors):The paper describes a straightforward set of logical experiments to explore the link between Mib1, Ryk and its role in CE. The primary conclusions of the paper are supported by data presented. Nevertheless, many questions remain unanswered. It remains unclear to me why the ta52b allele has such a minimal effect, the authors could elaborate on this. It also remains unclear how Ryk endocytosis contributes to effective PCP or how reduced Ryk endocytosis alters the organization or function of other PCP components. However, it is reasonable for these questions to be addressed in future studies.The authors have provided information about relevant data, reagents pertaining to eLife policy.

We thank the reviewer for his positive assessment of our work and agree that the fact that the *mib1^ta52b^* mutation dirsupts Notch but not PCP signaling is particularly puzzling. In the revised version of our manuscript we now provide experimental evidence that, like wild-type Mib1, Mib1^ta52b^ is indeed still able to promote Ryk endocytosis The corresponding data are displayed in the newly included Figure 3—figure supplement 2 and mentioned in the Results section:

“If Mib1-dependent Ryk endocytosis is important for CE, this process should be unaffected by the *mib1^ta52b^* mutation that disrupts Notch signaling (Itoh et al., 2003) but has no effect on gastrulation movements (Figure 1D). In accordance with this hypothesis, Mib^ta52b^ misexpression promotes a relocalization of Ryk from the plasma membrane towards intracellular compartments (Figure 3—figure supplement 2A,B), similar to the effect observed upon overexpression of wild-type Mib1 (Figure 3—figure supplement 1B,C).”

While these new experimental data provide a mechanistic explanation for the observations that 1, *mib1^ta52b^* mutants do not present CE defects (Figure 1D) and that 2, *mib1^ta52b^* RNA injection allows to rescue *mib1* morphant CE defects (Figure 1F), it remains to be established how the ta52b mutation specifically disrupts Notch but not PCP signaling. In the Discussion section we provide a potential explanation for this observation, but it is entirely clear that addressing this issue will require further studies that extend beyond the scope of the present work:

“These observations suggest that the capacity of Mib1 to mediate substrate ubiquitination is required for the PCP-dependent control of CE movements, but raise the question how the *mib1^ta52b^* mutation may specifically impair Notch but not PCP signaling? […] A possible – but currently entirely speculative – explanation for our observation that *mib1^ta52b^* mutants present impaired Notch but intact PCP signaling could be that the ta52b RF3 point mutation disrupts the interaction of Mib1 with a specific E2 enzyme required for Notch signaling, while having no effect on a distinct E2 enzyme involved in PCP.”

Reviewer #2 (Recommendations for the authors):The authors make the conclusion that the different phenotypes between ta52b and the proposed null alleles support a novel role for mib in regulating CE. While the authors expand on in the discussion, it is unclear to this reviewer if the ta52b is notch-specific or a hypo-morphic mutation.

We have now included new text in the Results section to further clarify this issue:

“The *mib1^ta52b^* mutation in the C-terminal Mib1 RING finger domain (RF3, Figure 1C) disrupts the ability of the protein to promote Δ ubiquitination (Itoh et al., 2003; Sharma et al., 2019; Zhang, Li, and Jiang, 2007). […] In spite of the strong, antimorphic Notch loss of function phenotypes observed in *mib1^ta52b^* mutants, axial extension occurs normally in these animals (Figure 1D, Figure 1—figure supplement 2A).”

The ΔRF123 is used as a dominant negative for Mib function. Better characterization of this reagent would be useful.

The Mib1∆RF123 protein variant has been characterized in the context of a previous study (Zhang, Li and Jiang 2007). We have now included additional text in the Results section to better explain the mode of action of this construct:

“All Mib1 functions known to date require its E3 ubiquitin ligase activity that is dependent of the presence of C-terminal RING finger domains (Guo et al., 2016). […] In the context of Δ/Notch signaling, truncated Mib1 variants that lack all three RING Finger domains sequester Δ ligands without promoting their ubiquitination and thereby exert a dominant-negative effect.”

A comment in the manuscript that needs to be reconciled. The author state how the mib morphants have the most severed CE compared to the putative null alleles, yet they use the anti-morphic Δ-ring construct to further deplete mib function. This is of concern to this reviewer.

We have now added additional text to further clarify this issue:

“The enhanced CE defects of Mib1∆RF123-injected *mib1* morphants are likely due to its capacity to interfere with maternally provided Mib1 protein which is unaffected by our morpholino that only impairs the splicing of zygotically produced *mib1* transcripts.”

We also further clarify this issue in the Discussion section:

“The partial loss of Ryk-positive endosomes that is observed in *mib1* mutants is most likely due to the perdurance of maternally deposited products. […] Accordingly, the most severe inhibitions of Ryk endocytosis are observed in embryos that have been co-injected with *mib1* splice morpholino and dominant-negative Mib1∆RF123.”

Reviewer #3 (Recommendations for the authors):The title 'The E3 Ubiquitin Ligase Mindbomb1 controls zebrafish Planar Cell Polarity' is misleading. There is no evidence that Mib1 regulates PCP in processes other than CE, independently of Notch signaling. Strictly speaking, Mib1 regulates endocytosis of Ryk, which modulates PCP during CE. Considering the fact that Mib1 also modulates Wnt/b-catenin signaling through Ryk (Berndt, 2011), it is too strong to state in the abstract that 'We show that Mib1, a known Notch signaling regulator, is also an essential PCP pathway component'. It is likely that Ryk is a context-dependent modulator of PCP rather than the PCP component, for example, by modulating Vangl2 (Macheda, 2012).

We agree that our current study provides evidence for an implication of Mib1 in the PCP-dependent regulation of gastrulation stage convergence extension movements but does not allow to establish Mib1 as a general PCP regulator. To avoid overinterpretation of our data, we have changed the title of our manuscript to *“*The E3 Ubiquitin Ligase Mindbomb1 controls PCP-dependent Convergent Extension movements during zebrafish gastrulation”.

Throughout the manuscript we have rephrased the text to clarify that our study establishes Mib1 as a regulator of PCP-dependent CE movements. In the Discussion section, we moreover specifically acknowledge that an important objective for future work will be to determine if Mib1 is involved for PCP signaling in other biological contexts:

“A major open question for future studies will be to determine whether Mib1 is also required for the control of PCP-dependent processes in other biological contexts. […] In this context, the analysis of gastrulation stage CE movements, which are critically dependent on PCP signaling but do not require Notch, provided a unique opportunity to unambiguously establish a function of Mib1 in the control of PCP-dependent morphogenetic movements. ”

Since maternal Ryk modulates PCP-dependent CE, one possible scenario is that the Wnt5b-Ryk axis regulates CE, which is dependent on Mib1. Several lines of evidence supports this; (i) MZwnt5b/ppt embryos show CE defects (Kilian, 2005), (ii) Wnt5b mediates Ryk endocytosis (Lin, 2010).My suggestion is to examine (i) whether Mib1 genetically interacts with wnt5b/ppt (using MO) and (ii) Wnt5 induced Ryk endocytosis is suppressed in mib1 null embryos expressing a dominant negative Mib1 (Mib^ΔRF123^).

In the revised version of our manuscript, we have now introduced a number of new experiments to address the functional relationship between Mib1 and Wnt5b/Ryk signaling.

In a first series of experiments, we introduced a previously validated wnt5b morpholino into Mib1 depleted (*mib1* morphant or *mib1^tfi91^* mutant) animals. In accordance with a functional relationship between Mib1 and Wnt5b-mediated PCP signaling, we observed that a dose of wnt5b morpholino that has no effect in WT controls is sufficient to generate and/or enhance CE defects in Mib1-depleted animals. The corresponding experiments are displayed in the newly included Figure 5 and presented in the Results section:

“Previous work in zebrafish showed that Ryk interacts with the PCP ligand Wnt5b to control gastrulation movements (Lin et al., 2010). […] In accordance with the previously reported function of Wnt5b/Ryk signaling in zebrafish gastrulation (Lin et al., 2010), our findings therefore suggest that through its ability to control Ryk localization, Mib1 may act as a modulator of Wnt5b-dependent CE movements.”

In a second series of experiments, we attempted to determine if Mib1 would be required for Wnt5b-mediated Ryk internalization. Unexpectedly, these experiments were however hampered by the fact that, in our hands and despite repeated prolonged efforts and the use of a large number of different experimental conditions, the misexpression of Wnt5b failed to promote a relocalization of Ryk from the plasma membrane to intracellular compartments similar to the one observed upon Mib1 overexpression.

To exclude that our failure to detect Wnt5b-driven Ryk accumulation in endocytic compartment was due to specific experimental conditions, we repeated these experiments using a wide range of concentrations of different Wnt5b (Wnt5b, Wnt5b-Myc, Wnt5b-IRES-CAAXGFP, Wnt5b-IRES-NLS-GFP) and Ryk (Ryk-GFP, Ryk-Myc, Ryk-HA, Flag-Ryk-Myc) constructs that were either obtained from previous studies or generated in the course of the present work. Neither the use of these different constructs, nor the use of different misexpression protocols (ubiquitous or localized Wnt5b/Ryk coexpression, Wnt5b and Ryk expression in adjacent cells, transplantation of Ryk-expressing cells into Wnt5b-expressing hosts), nor the use of different antibody staining protocols (PIPES or PBS-based fixation/staining buffers) allowed to provide convincing evidence for Wnt5b-mediated Ryk internalization. While we very occasionally (i.e. few cells of one or two embryos) observed an increase in intracellular Ryk accumulation, we simply never observed a clear relocalization of Ryk from the cell membrane towards intracellular compartments comparable to the one observed upon Mib1 overexpression.

Only at very high doses of Wnt5b RNA (300 pg) we observed a highly penetrant overall decrease of Ryk-GFP levels (Figure 5—figure supplement 1A,B), a phenotype that can also be observed upon overexpression of high amounts of Mib1 (Figure 3—figure supplement 1F). To determine whether the reduction of Ryk levels that is observed upon misexpression of high amounts of Wnt5b is dependent on Mib1 function, we repeated this experiment in *mib1^tfi91^* mutants that were additionally injected with RNA encoding the dominant-negative Mib1 variant Mib1∆RF123. Mib1 loss of function did not impair the Wnt5b-mediated downregulation of Ryk-GFP levels (Figure 5—figure supplement 1C,D), suggesting thereby that Mib1 is not important for the Wnt5b-mediated control of Ryk distribution.

The corresponding data have been integrated in the newly added Figure 5—figure supplement 1 and mentioned in the Results section:

“Mib1 could modulate Wnt5b/Ryk signaling by controlling overall cellular Ryk distribution or promoting a specific, ligand-dependent internalization of Ryk upon Wnt5b stimulation. […] In accordance with this hypothesis, the loss of Ryk-GFP expression that is observed upon overexpression of high amounts of Wnt5b does not require Mib1 function (Figure 5—figure supplement 1C,D).”